# Bromodomain factor 5 is an essential regulator of transcription in *Leishmania*

Nathaniel G. Jones [1✉], Vincent Geoghegan[1], Gareth Moore[1], Juliana B. T. Carnielli [1], Katherine Newling [1], Félix Calderón [2], Raquel Gabarró[2], Julio Martín[2], Rab K. Prinjha [3], Inmaculada Rioja[3], Anthony J. Wilkinson [4] & Jeremy C. Mottram [1]

*Leishmania* are unicellular parasites that cause human and animal diseases. Like other kinetoplastids, they possess large transcriptional start regions (TSRs) which are defined by histone variants and histone lysine acetylation. Cellular interpretation of these chromatin marks is not well understood. Eight bromodomain factors, the reader modules for acetyl-lysine, are found across *Leishmania* genomes. Using *L. mexicana*, Cas9-driven gene deletions indicate that BDF1–5 are essential for promastigotes. Dimerisable, split Cre recombinase (DiCre)-inducible gene deletion of *BDF5* show it is essential for both promastigotes and murine infection. ChIP-seq identifies BDF5 as enriched at TSRs. XL-BioID proximity proteomics shows the BDF5 landscape is enriched for BDFs, HAT2, proteins involved in transcriptional activity, and RNA processing; revealing a Conserved Regulators of Kinetoplastid Transcription (CRKT) Complex. Inducible deletion of BDF5 causes global reduction in RNA polymerase II transcription. Our results indicate the requirement of *Leishmania* to interpret histone acetylation marks through the bromodomain-enriched CRKT complex for normal gene expression and cellular viability.

[1] York Biomedical Research Institute, Department of Biology, University of York, York, UK. [2] GSK Global Health, Tres Cantos, 28760 Madrid, Spain. [3] Immunology Research Unit, Research, R&D GSK, Gunnels Wood Road, Stevenage, Herts SG1 2NY, UK. [4] York Biomedical Research Institute and York Structural Biology Laboratory, Department of Chemistry, University of York, York, UK. ✉email: nathaniel.jones@york.ac.uk

Gene transcription in eukaryotic cells is a complex process with layered regulation[1]. Post-translational modifications (PTMs) of histones in nucleosomes can encode an extra layer of information into chromatin, modifying transcriptional activity and leading to differential gene expression. Histone lysine acetylation is one predominant modification; it is interpreted by 'reader' proteins called bromodomains. In eukaryotic pathogens such as *Leishmania*, an organism with an unusual genome arrangement, the impact of histone lysine acetylation on transcriptional regulation is not well understood.

Bromodomains consist of 100–110 amino acid residues folded into a bundle of 4 helices connected by two prominent loops that form a hydrophobic pocket that can bind acetyl-lysine modified peptides. Conserved tyrosine and asparagine residues serve as acetyl-lysine recognition elements along with a network of water molecules in the pocket[2–4]. Bromodomains typically recognise acetyl-lysine residues of histone tails. Proteins containing bromodomains are often called bromodomain factors (BDFs). Other accessory domains in the BDF or its binding partners can then carry out other functions such as applying additional PTMs or chromatin remodelling. BDFs can regulate processes at specific regions of genomes such as promoters or enhancers, influencing differential gene expression, leading to cellular proliferation or differentiation. Inhibitors of these interactions have been intensely explored to identify pharmacological interventions for diseases driven by dysregulated BDF-driven processes[3–6].

Bromodomain proteins are poorly studied in kinetoplastid species, such as *Leishmania*, the causative agents of multiple human and animal diseases. Visceral leishmaniasis infects 50,000–90,000 people per year and the cutaneous forms of the disease, including those caused by *L. mexicana*, afflict up to 1 million people per year[7]. Kinetoplastids are deeply branched eukaryotes and their gene expression is radically different to the human host[8]. Genes are arranged into unidirectional polycistronic transcription units (PTU) of hundreds of non-functionally linked genes, and expression is driven from poorly defined transcriptional start regions (TSRs) that are often thousands of bases long[9]. The PTUs can run on either the plus or minus strand from transcriptional start regions at divergent strand switch regions[10]. Where PTUs then meet, a convergent SSR occurs, these are transcriptional termination sites (TTS). During transcription of protein-coding genes, RNA polymerase II (pol II) generates polycistronic pre-mRNAs that are processed by co-transcriptional cleavage, polyadenylation and trans-splicing events to produce mature mRNAs. Expression levels of individual genes are then typically regulated by the 3′ UTR, which is targeted by RNA binding proteins for stabilisation, sequestration or degradation[11]. In *Leishmania*, some highly expressed genes are found in tandem arrays or on supernumerary chromosomes to increase gene dose[12]. *Leishmania* also exhibits high levels of mosaic aneuploidy in cell populations as an adaptive survival strategy allowing plastic variation in gene dose[13]. Consequently, transcription of protein-coding genes by RNA polymerase II was thought to be constitutive with no sequence-defined, promoter-specific regulation[14]. However, it appears that cellular demarcation of transcriptional start regions might be mediated through histone acetylation to provide a platform of accessible chromatin. This suggestion comes from the identification of H3 acetylation at TSRs in *L. major*[14]. Histone acetyltransferases have been identified as essential for *L. donovani* survival and linked to specific histone modifications[15–18]. However, the way histone lysine acetylation influences transcriptional regulation is not well understood.

In this work, we validate five bromodomains factors as essential in *Leishmania* and characterise the biology co-ordinated by the essential bromodomain factor, BDF5. By applying inducible gene deletion, we establish the requirement for BDF5 in both cell culture and a mammalian host. We apply multiple -omics tools to characterise the function of BDF5, in particular using ChIP-seq to define the genomic distribution of BDF5, proximity proteomics to determine the processes occurring in BDF5-enriched loci and RNA-seq to explore the role of BDF5 in gene expression. Integrating the findings of these approaches we define BDF5 as an essential factor required for pol II transcriptional activity in *Leishmania*.

## Results

**Bromodomain factors in Leishmania.** Although bromodomain factors BDF1–5 were readily identifiable in *Leishmania* genomes[19], further PFAM and HMM searching identified another two potential bromodomain-containing proteins BDF6 and BDF7 (Fig. 1a, Table 1)[20]. BDF1–5 have identifiable tyrosine and asparagine residues in positions consistent with the conserved residues important for peptide binding in canonical bromodomains. BDF1–3 are small proteins <500 residues and contain a single bromodomain and no other identifiable domains. BDF4 is a larger protein with a centrally located bromodomain followed by a predicted CW-type zinc finger. BDF5 is the only *Leishmania* BDF that has tandem bromodomains. We refer to these as BD5.1 and BD5.2 and both are located in the N-terminal half of the protein. A more sensitive HHPred[21] analysis suggested remote structural homology to an MRG domain-like region (MORF4 -mortality factor on chromosome 4 -related gene) in the C-terminal half of the BDF5 protein. MRG domains can bind transcriptional regulators and chromatin remodelling factors[22–24]. BDF6 has a C-terminal bromodomain and is predicted to have an N-terminal signal peptide when analysed using SignalP4.1. BDF7 is the largest of the BDFs and contains a bromodomain in the C-terminal region of the protein preceded by a predicted ATPase and AAA domain. The bromodomain does not appear to have the conserved tyrosine and asparagine residues that are important for acetyl-lysine binding and may be non-canonical bromodomains or pseudo-bromodomains[25]. However, by HMMER analysis and alignment, it appears that BDF7 might be a homologue of the ATAD2 factor found in many other eukaryotes[26–28]. All of the predicted BDFs, apart from BDF6, contain K[K/R]x[K/R] motifs that can act as a monopartite nuclear localisation signal (NLS)[29]. The BDFs of *Leishmania* have orthologs in all other identified parasitic and free-living kinetoplastid organisms, with the exception of BDF1 in *Bodo saltans*[30,31]. BDF8 was later identified by HHPred analysis of a hypothetical gene identified using BDF5-proximity proteomics (this study), it may represent another pseudo-bromodomain.

To assess the essentiality of the seven BDFs in *Leishmania* promastigotes we used Cas9-targeted gene deletion. sgRNAs and repair templates were generated to target the CDS of each gene and replace it completely (Supplementary Fig. 1a)[32]. Two independent experiments were performed, using either blasticidin (*BSD*) or *BSD* and neomycin (*NEO*) drug resistance repair cassettes, leading to three semi-independent selections. Consistently, BDF6 and BDF7 null mutants could be isolated (Fig. 1b, Supplementary Fig. 1b–d). For BDF1–5 only heterozygote mutants were ever isolated, indicating that although the Cas9 system was functioning, a copy of the gene was required for promastigote survival and thus null mutants could not be generated.

LmxBDF5, while distinct from BET bromodomains, shares a characteristic tandem bromodomain arrangement reminiscent of human BRD2 and BRD4, or the yeast RSC4 protein[4]. These proteins have all been shown to play roles in regulating transcription, an interesting feature that prompted us to

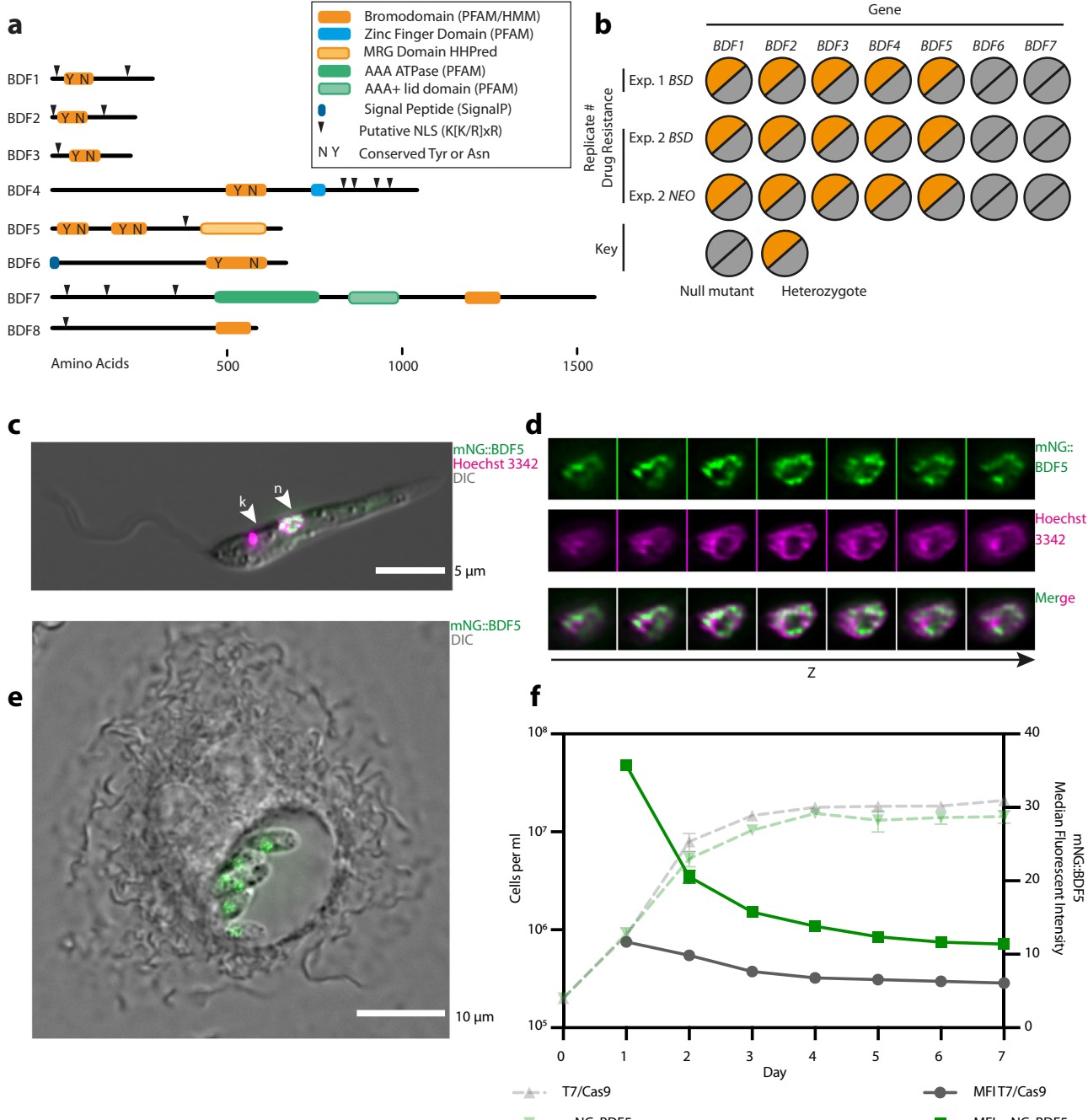

**Fig. 1 Overview of putative *Leishmania* bromodomain factors. a** Schematic showing protein domain architecture of *Leishmania* BDFs. **b** Overview of Cas9 gene deletion attempts of *BDF1–7* in *L. mexicana T7/Cas9* promastigotes. Two independent transfections were carried out, one using only *BSD*, another using both *BSD* or *NEO* as a drug selectable markers. For the *BSD* plus *NEO* experiment clones were selected using blasticidin and G418 in combination, or individually. No clones were recovered from the dual selection, but null mutants of BDF6 and BDF7 were recovered on all 3 occasions when a single drug selection was applied. No null mutants of BDF1–5 were ever recovered. **c** Live-cell fluorescent microscopy of *L. mexicana* promastigote expressing mNG::BDF5. Nucleus is denoted by arrowhead labelled n, the kinetoplastid DNA is indicated by arrowhead labelled k. **d** Channel separated Z-slices of the nucleus from the cell in (**c**). **e** Live-cell fluorescent microscopy of intramacrophage *L. mexicana* amastigotes expressing mNG::BDF5 endogenously tagged protein. **f** Expression levels of mNG::BDF5 during promastigote growth, determined by mNG signal in individual cells by flow cytometry. Dashed points denote mean cell density, error bars ± standard deviation, solid points denote median fluorescence intensity, $N = 3,20,000$ events per sample.

investigate BDF5 in greater detail. BDF5 homologs are identifiable in all the kinetoplastid genomes available in TriTrypDB. The level of amino acid conservation across the first bromodomain (BD5.1) is higher than the second (BD5.2), but in all cases the conserved tyrosine and asparagine residues are retained in both bromodomains (Supplementary Fig. 2a), these correspond to Y40, N90, Y201 and N247 in LmxBDF5. Both bromodomains have

X-ray crystal structures available in the PDB (PDB ID: 5TCM, 5TCK), verifying the bromodomain structural fold and confirming the positions of the conserved residues (Supplementary Fig. 2b). BDF5 was endogenously tagged using a Cas9-targeted approach to append a 3xMyc epitope and the green fluorescent protein mNeonGreen to the N-terminus[32] to generate *mNG::BDF5*. This modification preserves the 3′ UTR, which is

**Table 1 Gene IDs for *L. mexicana* BDFs and orthologues in selected trypanosomatids.**

| Name | *L. mexicana* | *L. donovani* | PDB ID | *T. brucei* | PDB ID | *T. cruzi* | PDB ID |
|------|---------------|---------------|--------|-------------|--------|------------|--------|
| BDF1 | LmxM.36.6880 | LdBPK_367210.1 | | Tb927.10.8150 | 5KO4 | TcCLB.506247.80 | |
| BDF2 | LmxM.36.2980 | LdBPK_363130.1 | 5C4Q | Tb927.10.7420 | 4PKL, 5CZG, 2N9G | TcCLB.507769.30 | 6NP7, 6NIM |
| BDF3 | LmxM.36.3360 | LdBPK_363520.1 | 5FEA | Tb927.11.10070 | 5C8G | TcCLB.509747.110 | |
| BDF4 | LmxM.14.0360 | LdBPK_140360.1 | | Tb927.7.4380 | | TcCLB.508857.150 | |
| BDF5 | LmxM.09.1260 | LdBPK_091320.1 | 5TCM, 5TCK | Tb927.11.13400 | 5K29, 6NEZ | TcCLB.510359.130 | 6NEY |
| BDF6* | LmxM.12.0430 | LdBPK_120390.1 | | Tb927.1.3400 | | TcCLB.510889.330 | |
| BDF7* | LmxM.11.0910 | LdBPK_110910.1 | | Tb927.11.6350 | | TcCLB.506297.110 | |
| BDF8* | LmxM.33.2300 | LdBPK_342070.1 | | Tb927.4.2340 | | TcCLB.506559.310 | |

PDB identifiers are provided for available structures. Proteins marked with * may be non-canonical or pseudo-bromodomains.

necessary for regulating endogenous mRNA levels in *Leishmania*, preserving native expression levels through growth and lifecycle stages. Live-cell widefield deconvolution microscopy of promastigotes identified that mNG::BDF5 localised to the nucleus (Fig. 1c). The distribution of mNG::BDF5 within the nucleus was heterogeneous, with foci found around the periphery of the nucleus and excluded from the nucleolus (Fig. 1d). The expression of mNG::BDF5 persisted in amastigotes where it was visualised in a structure consistent with the nucleus of intramacrophage amastigotes (Fig. 1e). *BDF5* mRNA was previously reported to be constitutively expressed in both lifecycle stages[33]. A seven-day time course experiment was performed where mNG::BDF5 levels in individual cells were measured by flow cytometry to determine the levels of BDF5 during promastigote growth (Fig. 1f). mNG::BDF5 levels were highest in rapidly proliferating cells during the first few days of growth and declined as the cells approached the stationary phase. By day 7, mNG::BDF5 levels were reduced by >60% compared to day 2. mNG::BDF5 signal was not completely reduced to the levels of the parental control strain, suggesting a low level of BDF5 expression was retained.

**BDF5 is essential in promastigotes and for murine infection.** To gain a higher quality validation of *BDF5* essentiality[34] and to investigate the phenotypes resulting from loss of BDF5 in promastigotes, an inducible knockout strain was generated using the DiCre system[35,36] (Supplementary Fig. 3a, b). An *L. mexicana* strain expressing dimerisable, split Cre recombinase was modified to carry a single, 6xHA epitope-tagged allele of *BDF5* flanked by loxP sites giving *L.mx::DiCre Δbdf5::BDF5:6xHA^{flox}/BDF5*. The second copy of BDF5 was then deleted using a *HYG* resistance cassette giving the strain *L.mx::DiCreΔbdf5::HYG/Δbdf5::BDF5:6xHA^{flox}*, referred to as *BDF5::6xHA^{−/+flx}*. In the absence of rapamycin, this strain grew normally as per the parental DiCre strain. However, following the addition of rapamycin, there was a marked reduction in parasite proliferation (Fig. 2a). Rapamycin was added to cultures at 300 nM for 48 h at which point the cultures were diluted to $1 \times 10^5$ cells per ml. Rapamycin was then added at 100 nM to suppress escape mutants and the growth phenotype observed. At the 144 h time point, the rapamycin-treated flasks contained ~98% fewer cells than the controls. Rapamycin did not affect the parental DiCre strain, indicating that the effect was specific to the floxed strain where *BDF5* could be deleted. This phenotype was reproducible and observed in an independent, clonal cell line (Fig. 2a). PCR analysis of these populations at 72 h after rapamycin addition revealed that the *BDF5::6xHA^{flx}* allele had been excised (Fig. 2b). Some leaky excision of the *BDF5::6xHA^{flx}* allele was detectable in the untreated control samples. The levels of BDF5::6xHA protein at 72 h were assessed by western blot, revealing a 90% reduction in the rapamycin-treated sample compared to the control samples (Fig. 2c). Total protein Stain-Free technology was used to provide

loading controls, considering the potential for BDF5 deletion to impact on the expression of housekeeping proteins. To demonstrate that the deletion of *BDF5* was essential for cellular survival a clonogenic assay was applied to characterise the cells resulting from *BDF5* excision (Fig. 2d). A 98% reduction in survival of the *BDF5::6xHA^{−/+flx}* strain was observed when cloned in the presence of 100 nM rapamycin, moreover, those cells that survived retained the *BDF5::6xHA^{flox}* allele. Deletion of BDF5 did not introduce a specific cell-cycle defect, although induced cultures appeared to have a reduced number of G1 arrested cells at 72 h post-induction (Supplementary Fig. 3c). The proportion of non-viable cells in the cultures at this point was ~10% (Supplementary Fig. 3d), in combination with our other experiments this suggests that BDF5 deletion leads to a rapid cytostatic phenotype followed by eventual cell death.

To ensure the phenotype was specific to *BDF5* deletion and not due to off-target effects, an allele of *BDF5::GFP* was re-introduced to the *BDF5::6xHA^{−/+flx}* strain using the pNUS episome[37] (Supplementary Fig. 4) Clonal survival experiments were performed in media lacking drug selection for the episome allowing for its loss if it confers no selective advantage. Clonal survival of the BDF5-complementation strain was ~50% after rapamycin addition, this is 25-fold higher than the non-complemented, induced samples. While not 100% complementation it reflects the potential for parasites to lose the episome (Fig. 2d), demonstrating the requirement for BDF5 for cellular survival. This experimental approach also allowed us to explore the essentiality of the individual bromodomains by making point mutations at the conserved asparagine residues in each bromodomain, N90 and N257 in BD5.1 and BD5.2 respectively (Supplementary Fig. 4). These were mutated to phenylalanine in anticipation that the bulky sidechain would occlude any binding peptide from the hydrophobic pocket[2,38,39]. Clonal survival was restored to similar levels as those observed for the *pNUS BDF5* complementation strain by the *BDF5^{N90F}* and *BDF5^{N257F}* mutants (Fig. 2d), indicating either that these mutations were not disruptive, or that any disruption due to mutation of a single BDF5 BD was tolerated by the cell. Three attempts were made to generate double mutations in N90F/N257F but no viable populations of cells were isolated, suggesting the *BDF5^{N90F/N257F}* is not tolerated by the cells. In light of this, we used a DiCre inducible system[40] to flip-on expression of an extra *BDF5^{N90F/N257F}::GFP* mutant allele to look for dominant-negative phenotypes (Supplementary Fig. 5a, b). Promastigote cultures induced to express BDF5^{N90F/N257F}::GFP exhibited a significant growth defect (Supplementary Fig. 5c, d), whereas those induced to express the BDF5::GFP protein did not exhibit this phenotype. These experiments demonstrate that individually both bromodomains are redundant, but that together they are required for the essential function of BDF5.

The ability to use the DiCre strains to validate target genes in *Leishmania* amastigotes is restricted due to the toxicity of

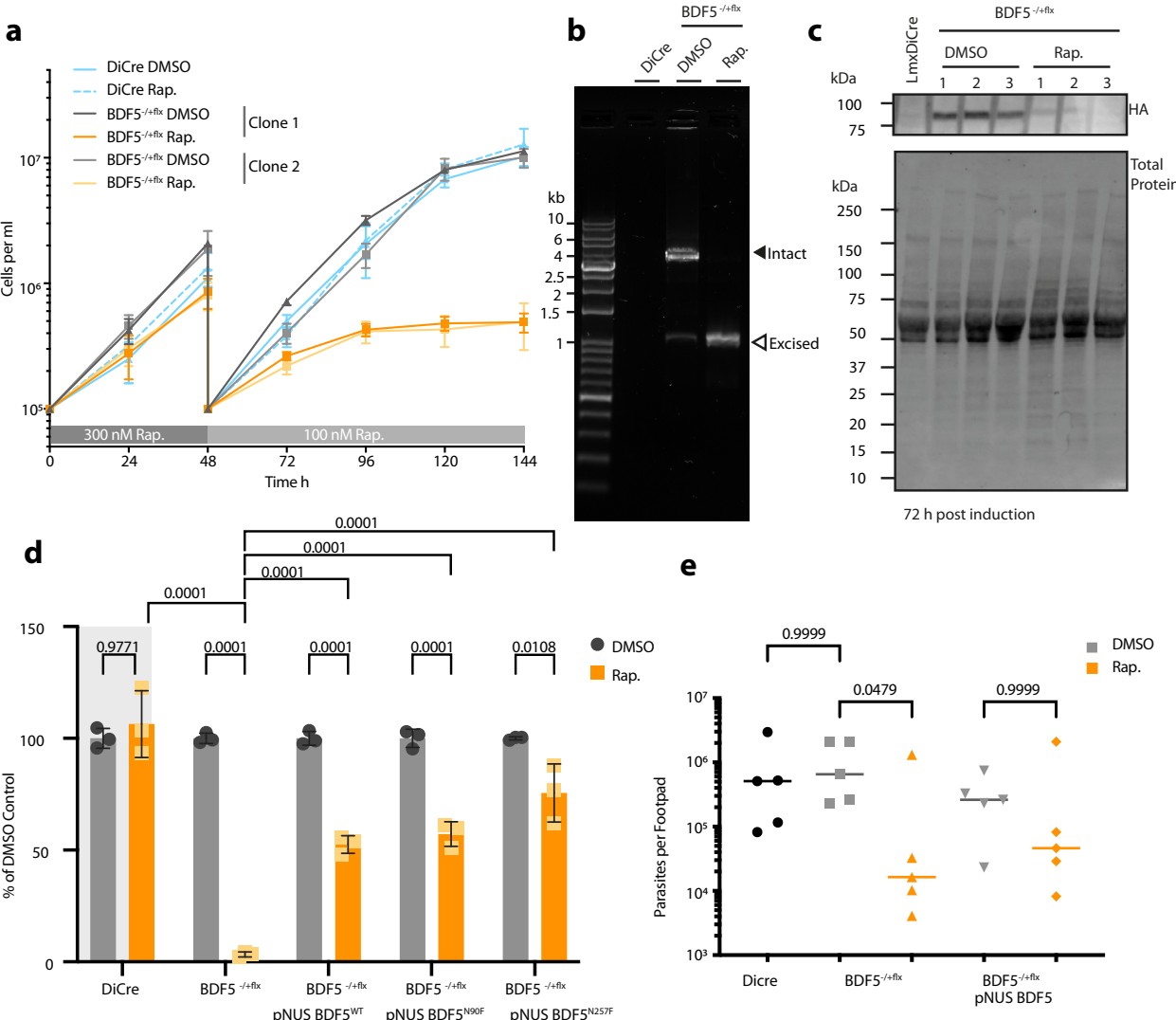

**Fig. 2 Characterisation of inducible knockout of BDF5 using DiCre. a** Growth curve of promastigotes treated with the inducing agent, rapamycin (Rap.), or the vehicle, DMSO. For the first 48 h 300 nM rapamycin was added. At 48 h the cultures were passaged, and the concentration of rapamycin was lowered to 100 nM. Daily counting was conducted of triplicate cultures, of two independent clones, using a haemocytometer. Points and error bars denote mean values ± standard deviation, $n = 3$. **b** PCR and agarose gel analysis of $BDF5::6xHA^{flx}$ gene excision at the 72 h timepoint in (**a**). Solid arrowhead denotes the intact $BDF5::6xHA^{flx}$ gene and open arrowhead denotes the excised locus after rapamycin addition. The DiCre lane indicates the lack of PCR product in the parental strain. **c** Western blot showing levels of BDF5::6xHA protein after 72 h of DMSO or rapamycin treatment, conducted in biological triplicate. **d** Results of clonogenic survival assay comparing BDF5-depleted cells with cell lines carrying episomal complementation of *BDF5* or mutated *BDF5* alleles. Bars denote the mean of the percentage clonal survival where each experiment was normalised to its own DMSO control. Error bars indicate standard deviation, values above are p values from 2-way ANOVA with multiple comparisons by Tukey's test, $N = 3$ replicate experiments. Lines denote comparisons performed by two-way ANOVA with associated *p*-values shown above. **e** Parasite burdens from infected mouse footpads determined by limiting dilution, individual points for each mouse with median values indicated by lines. Late-log cultures were pre-treated with 300 nM rapamycin and allowed to become stationary, prior to footpad infection for 8 weeks. Comparisons of Kruskal–Wallace test with Dunn's correction indicated with associated *p*-values written above, $n = 5$.

rapamycin to amastigotes and its immunomodulatory effect in mammals[35]. Therefore, mid-log promastigote cultures of *Lmx::DiCre*, $BDF5::6xHA^{-/+flx}$ or $BDF5::6xHA:^{-/+flx}:pNUS$ *BDF5::GFP* were treated with 500 nM rapamycin or DMSO for 72 h allowing them to induce deletion of BDF5 but still allow infectious, metacyclic promastigotes to accumulate in culture. Excision of the BDF5 gene was verified by PCR (Supplementary Fig. 6a) and the stationary cultures were used to infect BALB/c mice by a subcutaneous route into the rear footpad. No apparent differences were observed in the size of the resulting footpad lesions over the 8-week infection period (Supplementary Fig. 6b), however, there was a 50-fold reduction in the parasite burden of

the footpads when mice were infected with $BDF5::6xHA^{-/+flx}$ rapamycin-treated cells compared to the $BDF5::6xHA^{-/+flx}$ DMSO treated cells or the parental strain (Fig. 2e). The presence of the *pNUS BDF5::GFP* episome restored parasite burden in the rapamycin-treated strain to a level not significantly different to that observed in its uninduced control (Fig. 2e). The median parasite burdens of the $BDF5::6xHA^{-/+flx}$ rapamycin-treated strain in the popliteal lymph nodes was not statistically different (Supplementary Fig. 6c). DNA extracted from footpads and lymph nodes, including both host and amastigote DNA, was subjected to PCR analysis which detected non-excised $BDF5::6xHA^{-/+flx}$ consistent with BDF5 being essential for amastigote survival as

well as promastigote survival (Supplementary Fig. 6d). Clonal promastigote lines lacking the *BDF5::6xHA*$^{−/+flx}$ allele could only be derived from the populations containing an addback copy of BDF5 (Supplementary Fig. 6e). We conclude that BDF5 is essential for successful infection of the mammalian host and is likely to be essential for amastigote survival too.

**ChIP-seq reveals BDF5 genomic distribution**. Due to the importance of BDF5 for the survival of *Leishmania* parasites and the demonstration that it is a nuclear protein, we sought to identify where it might be found in the context of genomic architecture. The BDF5::6xHA protein expressed by the *BDF5::6xHA*$^{−/+flx}$ strain was analysed by chromatin immuno-precipitation sequencing (ChIP-Seq). We identified 175 regions where BDF5 was enriched on the genome; these peaks were distributed across all the 34 chromosomes and could be corre-lated with specific genomic features (Fig. 3, Fig. 4a, b). None of the BDF5-associated peaks were identified in control ChIP-seq experiments against the DiCre strain (Fig. 4a). Of the 175 total peaks, 56 (32%) were associated with TSRs in divergent strand switch regions (dSSRs). A further 30 (17%) were in subtelomeric regions likely to be transcriptional start regions based on the orientation of the polycistronic transcription unit. Forty-seven peaks (27%) were identified in internal regions of polycistronic transcription units (PTUs) and a further 11 peaks overlapped with isolated tRNA genes (6%). Intriguingly, 31 peaks (18%) were found at convergent strand switch regions, which are likely transcriptional termination sites. The size of the regions deter-mined to be enriched for BDF5 varied, with the mean of peak width found at dSSRs encompassing ~10 kb (Fig. 4c). The profile of BDF5 peaks over divergent strand switch regions tended to be broad and even, in contrast to the "twin-peaks" pattern seen for histone H3 acetylation in *L. major*[14] (Fig. 4a, d). Peaks at both divergent and convergent SSRs tended to be symmetrical although they were narrower and weaker at convergent SSRs (cSSRs) (Fig. 4d). Peaks found in PTUs were asymmetric, rising steeply to a peak with a shallow decay in the direction of the PTU transcription (Fig. 4e). The PTU peak enrichment levels were equivalent to those at dSSRs. The finding that BDF5 pre-dominantly localises to divergent SSRs and other TSRs suggests it plays a role in polymerase II transcription. However, as there was enrichment at a number of termination sites as well as at other classes of small RNA genes, BDF5 may play a more general role in a range of transcriptional processes. Therefore, we sought to analyse the protein complexes associated with BDF5 to give insight into its potential function.

**XL-BioID identifies the BDF5-proximal proteome**. To identify the functional properties of the environment proximal to BDF we applied an in-situ proximity labelling technique, crosslinking BioID (XL-BioID)[41]. The promiscuous biotin ligase BirA*, which generates a locally reactive (~10 nm) biotinoyl-5'-AMP[42], was fused to the N-terminus of BDF5 by endogenous tagging. The resultant parasites were incubated with 150 μM biotin for 18 h to permit labelling of proteins in proximity to BirA*::BDF5. The parasites were then treated with a limited amount of dithio-bis(succinimidyl propionate) (DSP) chemical cross-linker, to increase the capture of proximal proteins which were enriched with streptavidin, trypsin digested and processed for LC-MS/MS analysis. Importantly, a control cell line was treated in the same way to provide a control dataset of spatially segregated, nuclear proteins. The nuclear-localised protein kinase CLK2[43] (also called KKT19) was chosen as it is expressed at similar levels to BDF5 and localised to a distinct structure, the inner-kinetochore[44]. This provided a way to subtract common background proteins labelled

during the synthesis and trafficking of BDF5 to the nucleus as well as endogenously biotinylated cellular proteins. Following SAINTq interaction scoring, 156 proteins were determined to be enriched at 1% FDR (Fig. 5, Supplementary Data 2.). A set of 22 of these proteins were selected for endogenous tagging with 3xHA::mCherry in the mNG::3xMyc::BDF5 expressing strain to allow reciprocal co-immunoprecipitation and verification of the XL-BioID dataset (Supplementary Fig. 7a, b, Supplementary Data 1). This also showed there was no co-localisation of BDF5 and CLK2, further validating the latter as an appropriate control protein. Of the 22 proteins (excluding CLK2) we tested under crosslinking conditions 19 were found to co-precipitate BDF5.

The BDF5-proximal proteins were assessed for potential function (Fig. 5) and assembled into a loose network. We identified a core set of 11 highly enriched proteins (>10-fold), including the bait protein BDF5. Also identified were BDF3, BDF4 and Histone Acetyltransferase 2 (HAT2), along with several hypothetical proteins LmxM.35.2500, LmxM.08_29.1550, LmxM.12.0230, LmxM.33.2300, LmxM.24.0530. A component of the spliceosome LmxM.23.0650 was also identified, as was SUMO, which is likely conjugated to proteins in the interactome (SUMO is a PTM common in the nucleus[45,46]). BDF5 was enriched 35-fold compared to the control samples, BDF3 was enriched 41-fold and BDF4 26-fold. HAT2 was enriched 27-fold, strongly suggesting that these proteins are in very close proximity and that they may even form a stable complex. BDF6 (8-fold) and YEATS, a non-bromodomain acetyl-lysine reader, (4.8-fold) were also identified, consistent with the chromatin environment surrounding BDF5 being important sites of regulation through acetylation. The hypothetical proteins of the interactome were assessed by Phyre2 and HHPRED for remote structural homology to known domains that might indicate their function. LmxM.35.2500 was enriched >65-fold compared to the control samples. HHpred analysis detected remote homology to forkhead-associated domains, suggesting it may play a role in the recognition of phosphorylation sites. For LmxM.33.2300, which was enriched 22-fold, HHPRED searching detected remote structural homology related to bromodomains in the C-terminal region. However, it appears to lack the conserved asparagine and tyrosine residues. LmxM.33.2300 may represent a degenerate bromodomain, therefore we propose to name it BDF8. LmxM.24.0530, which was enriched almost 14-fold, is predicted to contain an EMSY N-Terminal Domain (ENT). EMSY is a protein implicated in DNA repair, transcription and human tumorigenesis[47,48]. LmxM.24.1230 was identified as 7-fold enriched, and domain searching identified a putative acetyltrans-ferase in the N-terminal region as well as PHD-Zinc Finger domain. The remaining 17 other enriched hypothetical proteins lacked structural homology to known protein domains.

Many proteins identified in the proximal proteome at lower enrichment levels play roles in processes associated with active transcription, broadly separated into RNA transcription and processing, including pre-mRNA cleavage, polyadenylation, splicing, cap-binding and quality control (nuclear exosome), indicating that these processes are occurring in proximity to BDF5. Components of RNA polymerase complexes were identified, including RPC4 associated with RNA polymerase III, RPB1, the largest subunit of RNA polymerase II and the RNA polymerase-associated protein LEO1. LEO1 is a component of the PAF1 complex, which plays numerous roles in transcriptional regulation. The basal transcription factors SNAP50, TFIIS-like protein (LmxM.32.2810), and a hypothetical protein (LmxM.22.0500) with remote homology to TFIIS helical bundle, ISW1 transcriptional elongator (LmxM.22.0500) were identified. Five components of the cleavage and polyadenylation specificity factor complex (CPSF) were identified and validated by reciprocal

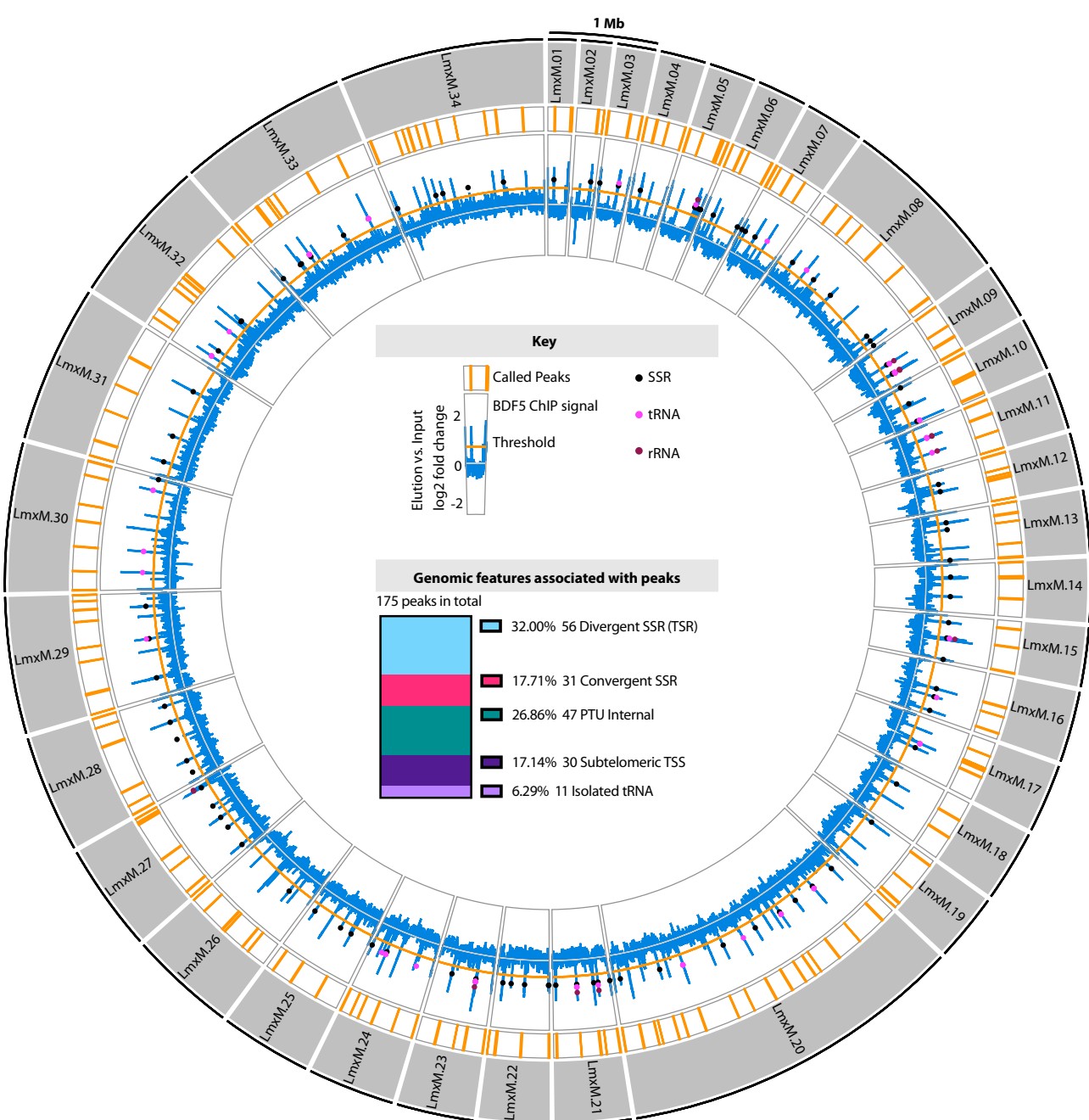

**Fig. 3 Genome-wide distribution of BDF5 determined by ChIP-seq analysis.** Outer circles: Circos plot representing the 32 MB *L. mexicana* genome. The 34 chromosomes are depicted by grey segments. Inner circles represent results from BDF5 ChIP-seq. Formaldehyde-crosslinked, enzymatically fragmented chromatin from the *BDF5::6xHA⁻/⁺ᶠˡˣ* strain was immunoprecipitated using anti-HA, reverse crosslinked and Illumina sequenced. Regions enriched >0.5 log2fold for BDF5 are indicated by the orange bars, the enrichment of BDF5::6xHA in the elution over the input chromatin is indicated in the blue line on a log2 fold scale, values are the mean derived from 3 ChIP replicates. Genomic features such as strand-switch regions (SSR), tRNA genes, and rRNA genes are indicated by coloured circles on this blue line. Inner panel: Key and stacked bar chart showing the genomic features associated with the peaks.

co-IP. Cleavage and polyadenylation occurs co-transcriptionally so this complex would be expected to be near polymerase complexes. Additionally, proteins associated with the splicing machinery of *Leishmania* were identified, including LmxM.23.065, a component of the spliceosome. Cap-binding proteins and members of the nuclear exosome were also identified, all indicative of the mRNA processing and quality control events that occur alongside transcription of the pre-mRNA. These hits are consistent with the ChIP-seq dataset and show BDF5 is located at sites of polymerase II transcriptional activity.

The ChIP-seq distribution of BDF5 identified it to be enriched not only at TSR regions but also at rRNA and tRNA genes and some polymerase II termination sites (cSSRs). It was interesting to discover proteins in the XL-BioID that are involved in the maturation of both tRNAs and rRNAs, placing BDF5 in proximity to the transcription and maturation of different classes of RNAs. Base J is associated with termination sites in *Leishmania*[49], and the base J-associated glucosyltransferase JBP1 (LmxM.36.2370) was found to be 3-fold enriched over the control, potentially indicating that BDF5 may occasionally be found at sites linked to transcriptional termination. Interestingly,

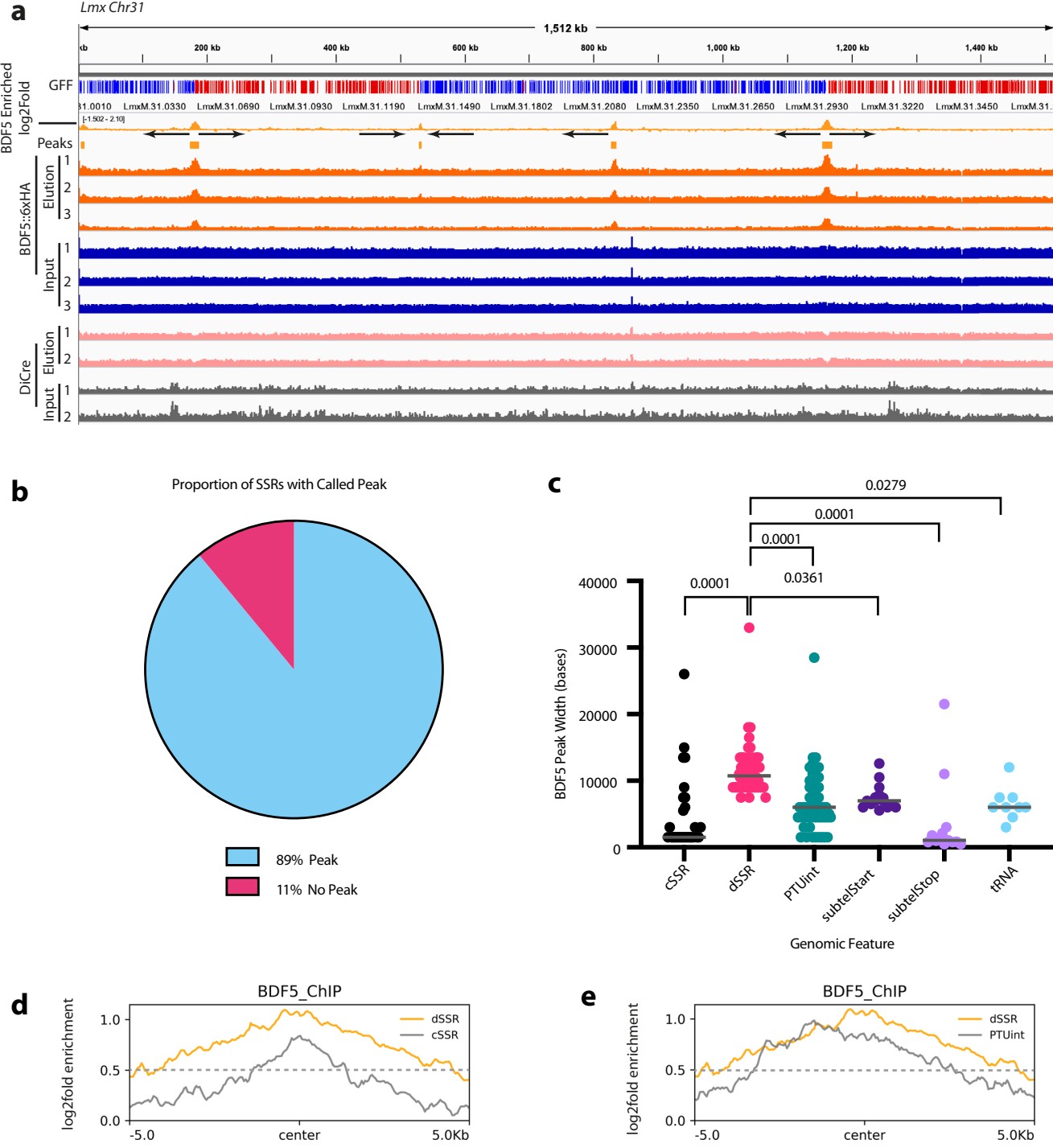

**Fig. 4 ChIP-seq analysis of BDF5 distribution on chromatin. a** IGV genome browser view using chromosome 31 as an example, indicating the genes in polycistronic transcription units (colour and arrow coded by direction), the read depth for input and eluted sample of the ChIP-seq of $BDF5^{-/+flx}$ ($N = 3$) and the enrichment of DNA associated with BDF5:6xHA on a log2 fold scale, GFF (gene feature file) indicates gene CDS coloured by strand (red +, blue -). Control ChIP-seqs were performed for the DiCre strain which does not contain a HA-tagged protein and input (grey tracks) and eluted (pink tracks) samples are represented below ($N = 2$). Peaks were defined as regions being >0.5 log2fold enriched in the elution versus input. **b** Pie chart indicating the proportion of SSRs with a BDF5-enriched peak. **c** BDF5 peak size at different genomic regions, cSSR (convergent strand switch region, $n = 28$), dSSR (divergent strand switch region, $n = 54$), PTUint (internal PTU peak, $n = 52$), subtelStart (subtelomeric peak consistent with PTU transcriptional start, $n = 15$), subtelStop (subtelomeric peak consistent with PTU transcriptional stop, $n = 13$), tRNA (tRNA gene located away from any of the other features, $n = 9$). Values above denote p-value from Kruskal–Wallace test to compare samples. **d** Metaplot of average BDF5 fold enrichment at dSSR ($n = 60$) and cSSR ($n = 40$) regions. **e** Metaplot of BDF5 average BDF5 fold enrichment at dSSR ($n = 60$) and PTU internal peaks ($n = 56$).

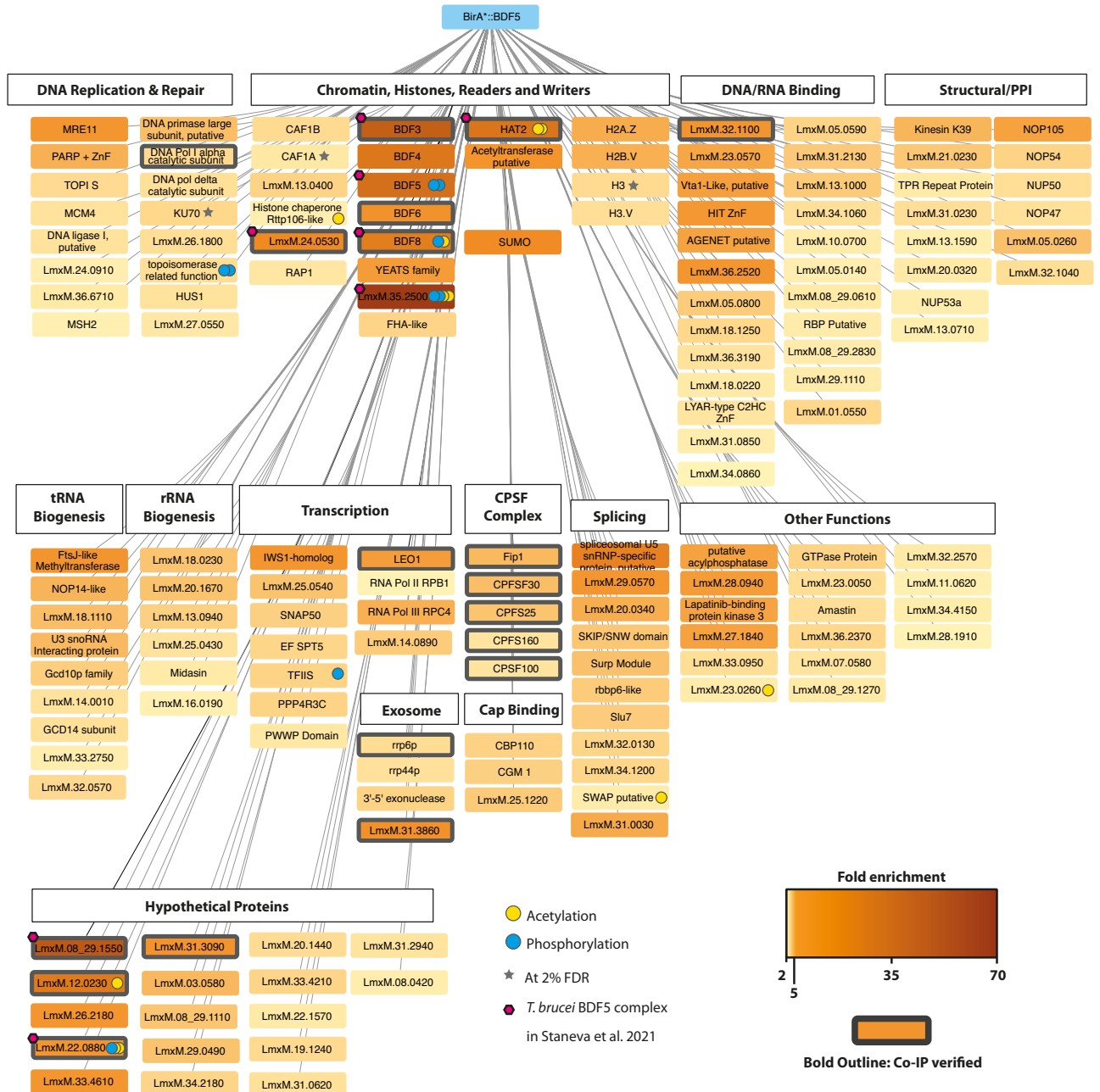

**Fig. 5 The BDF5-proximal proteome determined by XL-BioID.** Network indicates proteins determined to be spatially enriched in proximity to BirA*::BDF5 following in vivo biotin labelling and DSP crosslinking during the XL-BioID workflow. Biotinylated material was enriched using streptavidin magnetic beads, trypsin digested and analysed by LC-MS/MS. SAINTq analysis was performed to determine statistically significant enrichment scores. Fold enrichment values are encoded in the colour intensity of the protein boxes; these were calculated from label free protein intensities against a control expressing CLK2::BirA*, from 3 replicate experiments. A bold outline on a box indicates that BDF5 co-purified with this protein in a reciprocal co-immunoprecipitation experiment. If a post-translational modification (PTM) was detected for a protein, this is indicated using a coloured circle. Proteins are grouped by functional annotations or previously published data of complexes in *Leishmania* or *Trypanosoma*. Proteins represented are those identified at 1% false-discovery rate (FDR), those marked with a grey star denote those identified at 2% FDR for selected proteins. CPSF stands for Cleavage and Polyadenylation Specificity Factor complex, PPI stands for protein-protein interactions. Magenta hexagons indicate members of the BDF5, BDF3, HAT2 complex reported in *T. brucei*.

many factors associated with the detection and repair of DNA damage were also found in the proximal proteome, together with factors associated with DNA replication. It is known that DNA damage can occur in transcriptionally active regions due to the formation of RNA-DNA hybrids called R-loops[50]. Some of the origins of DNA replication in *Leishmania* coincide with transcriptional start regions, suggesting we can detect this association in the XL-BioID data[51].

Because the samples were trypsin digested, it was difficult to obtain information on histone tails. Nevertheless, we were able to detect some peptides from the core of histones and histone variants as significantly enriched in proximity to BDF5. Peptides were detected for H2A.Z, H2B.V, H3 and H3.V. In *T. brucei*, the H2A.Z and H2B.V variants have been localised to divergent SSRs[52], where H2A.Z plays a role in the correct positioning of transcription initiation. H2A.Z and H2B.V are also essential for

*Leishmania*[53]. H3.V localises to convergent SSRs in *T. brucei* and is not essential for *Leishmania*, nor does it play a role in transcriptional termination in this organism[53]. Mining the XL-BioID data further we were able to detect a number of acetylated peptides. Acetylation sites were detected on HAT2, BDF8, FHA-like protein (LmxM.35.2500), a putative Rttp106-like histone chaperone and several hypothetical proteins (Supplementary Data 2). However, we were again unable to detect acetylated peptides derived from histones.

To further explore the potential interaction partners of BDF5 a subset of immunoprecipitations under non-crosslinking conditions were conducted using epitope-tagged strains for LmxM.08_29.1550, LmxM.24.0530, LmxM.22.0880, HAT2, BDF3, BDF8, LEO1, and CLK2. As expected CLK2 did not co-precipitate BDF5, but all the other proteins did with the exception of BDF3 (Supplementary Figure 7c). The existence of stable interactions between these proteins and BDF5, likely represents a protein complex.

The capacity of XL-BioID to enrich large amounts of proximal material allows it to be combined with other methods, such as phosphoproteomics[41]. We engineered a cell line to carry BDF5::miniTurboID for faster labelling kinetics and higher temporal resolution, allowing us to explore BDF5-proximal phosphorylation events across the cell cycle of hydroxyurea synchronised cultures. Following synchronisation release, 30-minute biotinylation timepoints were carried out 0, 4 and 8 h corresponding to early-S, S and G2/M phase respectively. Samples were processed using the XL-BioID workflow, then proximal phosphopeptides were enriched using Ti-IMAC resin prior to LC-MS/MS analysis. The resulting dataset was compared to a reference phosphopeptide dataset derived from the kinetochore protein KKT3[41]. Two BDF5-proximal phosphopeptides were identified in early-S phase rising to 19 and 13 as the cells progressed through the S and G2/M phases respectively (Supplementary Fig. 8, Supplementary Data 3). Of these, 14 unambiguous phosphosites were detected for proteins in proximity to BDF5, including several for BDF5 itself, pS135, pS133, pS317 and pS330. S135 and S133 are located between the two bromodomains, while S317 and S330 are located after the second bromodomain. LmxM.35.2500, which was highly enriched in the original XL-BioID and identified to contain a putative FHA domain, was itself found phosphorylated at S202, S208 and S545. LmxM.33.2300 (BDF8) was found to contain an ambiguous phosphorylation at one of six sites in the region of S50-S61 (Supplementary Fig. 8). Despite the detection of multiple phosphosites, only a single protein kinase was identified in proximity to BDF5, LmxM.25.1520 (LBPK3), an orphan kinase with unknown function that has been reported to bind lapatinib[54].

The combined ChIP-Seq and XL-BioID data defined where BDF5 localises on the genome, as well as the surrounding protein landscape. This pointed towards a role for BDF5 in promoting transcriptional activity and provided a starting point to develop assays to characterise the phenotype of BDF5-induced-null strains.

**BDF5 depletion results in an RNA Polymerase II transcriptional defect.** As most of the BDF5-enriched regions of the genome corresponded to transcriptional start regions, and the proximal proteome contained factors associated with the transcription and maturation of various classes of RNA, we sought to assess the effect of BDF5 depletion on cellular RNA levels. Promastigote cultures were stained for total RNA content using SYTO RNASelect fluorescent stain at 24, 48 and 72 h timepoints and measured by flow cytometry (Fig. 6a). For the *Lmx::DiCre* strain, the addition of rapamycin caused no changes in the levels of total RNA staining. SYTO RNASelect staining increased as cells progressed through log phases of growth at 48–72 h time points.

However, once BDF5 was deleted from the *BDF5::6xHA*$^{-/+flx}$ cell line by the addition of rapamycin, there was a pronounced increase in the number of cellular events containing very low levels of RNA staining, such that a profile at 72 h overlaps with that for unstained control cells. This result suggested that total levels of transcription were reduced upon BDF5 deletion from cells. We investigated this in more detail by using total, stranded RNAseq that included External RNA Controls Consortium (ERCC) Spike-in controls[55]. Cultures of *BDF5::6xHA*$^{-/+flx}$ treated with Rapamycin or DMSO were harvested, then RNA extraction buffer spiked with the 92 synthetic ERCC RNAs was used to lyse the parasites for RNA purification. Following sequencing and read mapping these RNAs were used to provide a normalisation channel. Overall, a > 50% reduction in the median read depth was observed across protein-coding genes on all chromosomes (Fig. 6b). When normalised read depths were compared using metaplots of divergent SSRs, this ~50% reduction in transcriptional levels was reflected (Fig. 6c). However, no positional effects were observed on transcriptional start regions (Fig. 6d). The 50% reduction in read depth was reflected across PTUs (Fig. 6e) and at convergent SSRs (Fig. 6f). Strand-specific read depth at cSSRs did not indicate any increase in transcriptional readthrough in BDF5-induced-null cells (Fig. 6f), suggesting the BDF5 located at these termination sites is not playing a role in transcriptional termination. Overall, these results indicate BDF5 is important for global pol II-dependent gene transcription. Due to the rRNA depletion method used and the low coverage over tRNA genes, we could not assess if pol I or pol III transcripts levels were reduced. To address this, we conducted spike-in controlled RT-qPCR against 3 reporter transcripts for pol I (18 s rRNA), pol II (Cyclophilin A), and pol III (tRNA$^{Lys}$). This showed that after BDF5 depletion only the Cyclophilin A gene was significantly reduced in expression level (Fig. 6g), suggesting BDF5 plays a key role in RNA pol II mediated transcription but is not directly required for other RNA polymerase function.

Transcriptionally active regions of kinetoplastid genomes often accumulate DNA damage which occurs due to the formation of DNA-RNA hybrids (R-loops)[50,56]. As we detected proteins involved in co-ordinating DNA repair in the BDF5-proximal proteome, and that this appears to be a broader feature of BDF protein networks[57,58], we examined if there was a link between BDF5 and the DNA damage response in *Leishmania*. *BDF5*-induced-null promastigotes cease growing quickly, whereas parasites deficient for genome-stability factors often die slowly[40], suggesting maintaining genome integrity is not the primary role of BDF5. Indeed, after using western blotting to detect γH2A phosphorylation[59], a sensitive marker for the cellular response to DNA damage, we could not detect any increase in γH2A signal in BDF5-depleted cells, nor was there any detectable difference in the γH2A response of these cells to a non-specific DNA damaging agent, phleomycin (Supplementary Fig. 9). This indicates that there is no direct or secondary role for BDF5 in DNA damage response. Despite enrichment in the BDF5-proximal proteome for mRNA splicing factors, we did not find evidence to support trans- or cis-splicing defects in BDF5-induced-null mutants using a qualitative RT-PCR assay. A positive control strain of *L. mexicana* was generated using CRISPR/Cas9 precision editing of *CRK9* to change the codon for the gatekeeper methionine (M501) to a glycine codon (Supplementary Fig. 10)[60]. This mutant is specifically inhibited by the bumped kinase inhibitor 1NM-PP1 leading to defects in splicing. Cis-splicing of poly-A polymerase and trans-splicing of cyclophilin A mRNA was examined by an RT-PCR method that could detect the pre-mRNA and mature mRNA. CRK9$^{M501G}$ inhibition resulted in accumulation of unspliced pre-mRNAs but deletion of BDF5 did not, indicating no role for BDF5 regulation of splicing (Supplementary Fig. 11).

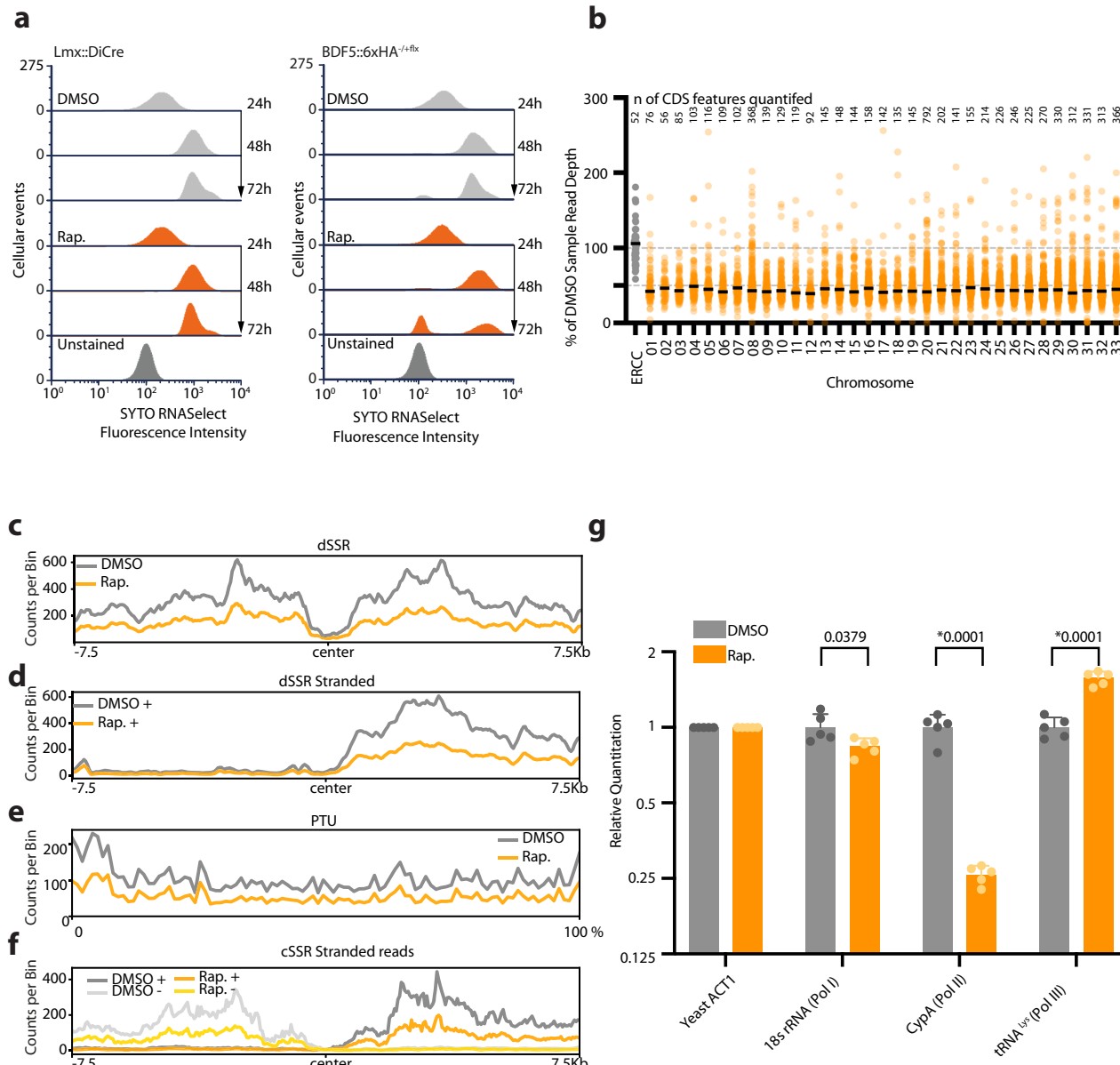

**Fig. 6 Effect of BDF5 depletion on RNA levels and gene expression. a** Flow cytometry of cells stained with SYTO RNASelect Stain to measure total RNA levels in *Lmx::DiCre* strains or the *BDF5*$^{-/+flx}$ strain treated with rapamycin or DMSO over a 72 h time course. 20,000 events measured per condition. **b** Dot plot of total RNA-seq reads per protein-coding gene scaled to ERCC spike-in controls, then as a percentage of the DMSO control sample, separated per chromosome, conducted at a 96 h timepoint. Black lines denote the median of the scaled response for each chromosome, individual data points are means of 2 separate RNA seq experiments, the number of CDS features quantified on each chromosome is indicated above the dot plots. **c** Metaplot of divergent SSR (*n* = 60) for DMSO treated or rapamycin-treated *BDF5*$^{-/+flx}$ showing combined reads from the positive and negative strands. **d**. Metaplot of reads mapping to the + strand, normalised to ERCC control at divergent SSRs (*n* = 60) of DMSO treated or rapamycin-treated *BDF5*$^{-/+flx}$ cultures. **e** Metaplot of + stranded RNA-seq reads normalised to ERCC spike-in controls for PTUs (*n* = 120), on a scale of 0–100%. **f** Metaplot of reads mapping to the + and − strands, normalised to ERCC control at convergent SSRs (*n* = 40) of DMSO treated or rapamycin-treated *BDF5*$^{-/+flx}$ cultures. Metaplot data is from 1 representative of the three replicate RNA-seq datasets. **g** Spike-in controlled SYBR RT-qPCR of reporter genes for Pol I, II, III. BDF5 deletion was induced for 96 h and total RNA was extracted with lysis buffer spiked with yeast total RNA to provide a normalisation channel using a primer set against yeast actin, allowing comparison of the relative 18s rRNA, Cyclophilin A, and tRNA$^{Lys}$ RNA levels compared to DMSO treated cells. Bars denote mean, error bars denote standard deviation. Comparisons by multiple two-sided *t* test, corrected with Benjamini and Hochberg method, p-values indicate above, * denotes a discovery, *n* = 5 replicate PCR reactions. ACT1 values were not compared as this was the normalisation target.

## Discussion

Kinetoplastid parasites have evolved a genomic architecture that requires most genes to be transcribed constitutively, with regulation being achieved post-transcriptionally or through specialised solutions such as altered gene dose[8]. Pol II transcriptional start regions may simply be maintained as open chromatin.

However, recent evidence has indicated these regions are actively regulated, particularly through histone acetylation. How the cell interprets these marks is not completely understood. Bromodomains are clearly critical components of this process in *Leishmania*; we were unable to generate null mutants in five of the seven bromodomain encoding genes, also implying there is no

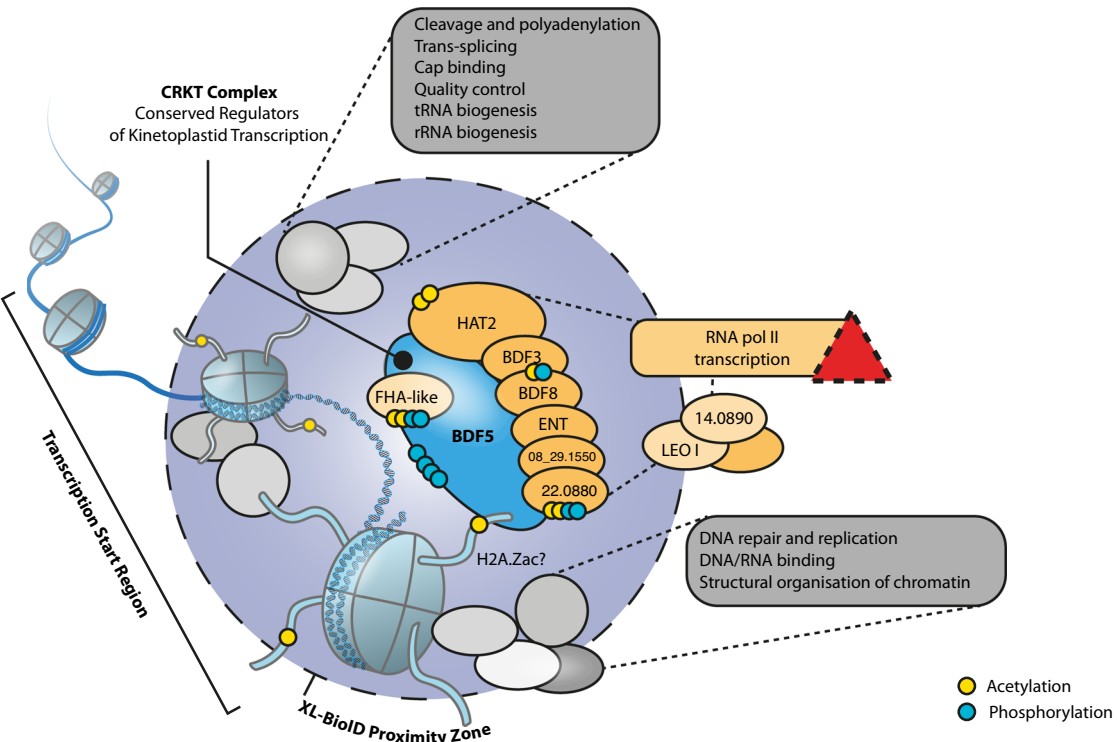

**Fig. 7 A proposed model of the BDF5-defined chromatin landscape and the CRKT complex.** BDF5 localised to chromatin with the CRKT complex members depicted as interacting directly and influencing transcription, either directly or through a putative PAF1-like complex. The red triangle indicates a disruption of transcription once BDF5 is deleted. Other proximal processes are indicated in grey boxes. Indications of detected acetylation and phosphorylation are depicted as yellow or blue dots respectively. Juxtaposition and stoichiometry of complex members are for illustration only, not to scale.

redundancy in their individual functions. Although failure to generate a null is the most basic standard of genetic evidence for essential genes[34], we were able to generate high-quality, genetic target validation for BDF5, using inducible DiCre both in the promastigote stage and during murine infections. BDF5 expression was confirmed in both stages and expression levels were correlated with cellular growth rate in promastigote stages. Combined with the rapid cytostatic phenotype occurring upon BDF5 inducible deletion, followed by cell death, this identifies BDF5 as a regulator of cell growth and survival. This finding demonstrates that the interpretation of histone acetylation is important for cellular survival (Fig. 7), although for *Leishmania* the specific histone PTMs found at TSRs are not currently defined.

Our starting hypothesis was that BDF5 would localise to polymerase II transcriptional start regions, so it was surprising to find BDF5-enriched peaks associating with other sites including rRNA genes, tRNA genes and convergent strand switch regions. This suggested BDF5 played a broader role in recruiting or regulating chromatin to promote transcription by multiple polymerase complexes. This was further emphasised by our proximity proteomics dataset, which revealed BDF5 to be close to proteins involved in different processes linked to transcriptionally active chromatin, in particular RNA maturation factors, DNA repair factors and polymerase-associated complexes. However, our phenotypic analyses appeared to rule out roles for BDF5 in influencing the DNA damage response, the cis- and trans-splicing of mRNA or transcription mediated by RNA pols I and III. Instead, they indicate that the role of BDF5 is exclusively in the normal transcription of protein-coding genes by pol II.

Spike-in controlled total RNA seq was previously used to study the influence of HAT1 and HAT2 on transcription in *T. brucei*[55]. It is striking that BDF5 knockout in *L. mexicana* phenocopies HAT1 knockdown in *T. brucei*, both resulting in an overall

reduction in transcription levels. In *T. brucei*, HAT1 is required for acetylation of H2A.Z and H2B.V. Depletion of HAT1, and thus H2A.Z acetylation levels, leads to a 10-fold decrease in the amount of chromatin-bound pol II, resulting in 50% reduced transcriptional activity. Intriguingly, pol II levels at TSRs were not affected; the authors suggested H2A.Z acetylation is required for optimal transcription by bound pol II. As BDF5 knockout phenocopies HAT1 depletion and results in lower pol II activity, it might therefore be involved in reading or applying acetylation of H2A.Z. Surprisingly, although we find HAT2 proximal to BDF5, HAT1 was neither enriched nor detected in our XL-BioID dataset. This suggests that there is a distinct spatial separation between BDF5, HAT2 and HAT1 in *Leishmania* (assuming there is no technical reason HAT1 cannot be labelled by XL-BioID). BDF5 co-precipitated with HAT2 in the absence of a covalent cross-linker (Supplementary Fig. 7), suggesting their complex is stable. We did not observe changes in the sites of transcription initiation, suggesting the HAT1/BDF5 phenotype over-rides any effect on HAT2 dysfunction. Purified *L. donovani* HAT2 has been shown to acetylate H4K10 and it appears to be essential as only heterozygotes can be generated using traditional knockout strategies[18]. *L. donovani* HAT2[−/+] heterozygotes grow slowly and display a cell-cycle defect. Transcription initiation at TSRs was not examined but the reduction of H4K10[ac] in a HAT2[−/+] background did not lead to global transcription reduction. However, the expression of *Cyclin 4* and *Cyclin 9* mRNAs was reduced in this mutant. The regions upstream of these genes were found to be enriched for H4K10[ac], and this was also reduced in the HAT2[−/+] strain. Intriguingly, transcription of these genes was cell-cycle dependent[18]. Future work could investigate the requirement of BDF5 for this cell-cycle dependent gene transcription in *Leishmania*, which is an interesting observation given the lack of obvious gene-specific promoters.

A recent immunoprecipitation dataset of *T. brucei* chromatin factors[61] defined a complex consisting of BDF5, BDF3, HAT2, BDF8 and orthologs of the hypothetical proteins LmxM.24.0530, LmxM.22.0880, LmxM.35.2500 and LmxM.08_29.1550; proteins that were highly enriched in our BDF5-proximity proteome (Fig. 5). We propose that these proteins represent a Conserved Regulators of Kinetoplastid Transcription (CRKT) Complex (Fig. 7), as orthologues are identified in many kinetoplastid species (Supplementary Data 1). CRKT is likely to be associated with transcriptional start regions with the multiple bromodomains potentially playing different roles. The association of BDF5 at transcriptional termination regions, albeit in lower amounts, could indicate that BDF5 is also part of a mobile complex that progresses along chromatin and accumulates at start and termination sites. One complex could be the PAF1 complex, a multifunctional complex associated with pol II initiation, elongation, pausing and termination[62]. However, the PAF1 complex is poorly characterised in kinetoplastids and the PAF1 protein itself lacks identifiable orthologs in these organisms. *T. brucei* BDF5 has been suggested as a potential component of a transcription initiation complex due to its dual bromodomains and the interaction with proteins homologous to TFIID TAF1 (which also contains 2 bromodomains as well as protein kinase and acetyltransferase activities).

Our findings further illustrate the power of proteomic approaches for studying chromatin regulation in kinetoplastids where the large TSRs allow plentiful material to be derived[55]. Combined with XL-BioID this allowed for the enrichment of PTMs to be determined for many of the complex members. The phosphorylation of the region connecting the two BDs might represent a site of PTM-dependent regulation of BDF5. Regulation of BDF function by phosphorylation has been reported, for example, phosphorylation of human BRD4 by CK2 adjacent to the second bromodomain allows this region to interact with a downstream basic residue-enriched interaction domain[63,64]. This interaction unmasks the second bromodomain and allosterically derestricts the first bromodomain allowing engagement with acetylated chromatin and site-specific transcriptional regulation[63,64]. This interaction has been reported to occur in *trans* thus allowing BRD4 to dimerise in a head-to-tail arrangement upon phosphorylation[65]. Intriguingly BDF5 has a basic patch (R406-R430) downstream of the phosphosites adjacent to BD5.2. The effect of phosphorylation on the oligomeric state of BDF5 and its regulation of transcriptional activity remains to be investigated. Adaptation of the XL-BioID workflow to include a chemical derivatisation step (e.g. stable isotope acylation) could allow it to be used to detect the normally trypsin-labile histone tail peptides in proximity to BDF5 or other BDFs, potentially identifying their native binding partners and providing locus-specific views of histone PTMs.

As bromodomains are chemically tractable targets it may be possible to develop specific inhibitors that target *Leishmania* BDF5. Such compounds would be of high value to the investigation of BDF5-dependent transcriptional regulation in kinetoplastids, allowing for precise temporal disruption of BDF5 and the processes that it coordinates. It should be noted that parasites expressing BDF5 with singly-mutated bromodomains were viable, requiring both to be disrupted to observe a reduction in parasite growth. Potential BDF5 inhibitors would likely require a bi-specific molecule, a PROTAC (proteolysis targeting chimera) molecule (not yet realised in kinetoplastids), or a mono-specific inhibitor that can perturb the complex enough to be fatal for the cell.

In summary, our findings identify the importance of the linkage between histone acetylation and transcriptional regulation by bromodomain factors in a eukaryote that is divergent from opisthokonts such as the humans host. Because of their unusual features, kinetoplastids are ideal organisms for investigating the evolution of chromatin regulation by acetylation.

## Methods

**Molecular Biology**. Computational sequence analysis, design of vectors, primers and PCR fragments was performed using CLC Main Workbench (Qiagen). Oligonucleotides were synthesised by Eurofins Genomics. High-fidelity PCRs were conducted using Q5 DNA polymerase (NEB) according to the manufacturer's instructions. Low-fidelity screening PCRs were conducted using Ultra Mix Red (PCR Biosystems) according to the manufacturer's instructions. Vectors for DiCre strain generation were generated as previously described[35] using Gateway Assembly (Thermo Fisher). PCR amplicons were resolved in 1% agarose (Melford) TBE gels containing 1x SYBRsafe and visualised on a Chemidoc MP (BioRad). A full list of oligonucleotides and vectors are presented in Supplementary Data 1. Sanger sequencing to verify plasmids etc. was conducted by Eurofins Genomics.

Protein samples of cells were generated by taking $2.5 \times 10^7$ log phase promastigotes, lysing in 40 μl LDS (lithium dodecyl sulfate) sample buffer supplemented to 250 mM DTT and heated to 60 °C for 10 min, after cooling, 1 μl of Basemuncher (Abcam) was added and the sample incubated at 37 °C to degrade DNA and RNA. Samples were separated in TGX Stain-Free SDS-PAGE Gels (BioRad) and the total protein labelled and visualised using a propriety trihalo compound activated by UV light in a BioRad ChemiDoc MP. Western blotting was performed using an iBlot II (Invitrogen) and the associated PVDF cassettes, using program P0. Membranes were blocked with 5% milk protein in 1x Tris Buffered Saline Tween-20 0.05%. Primary and secondary antibodies are listed in Supplementary Data 1 and were detected using appropriate fluorescent channels of chemiluminescent channels of a Chemidoc MP (BioRad), using Clarity Max Western ECL Substrate (BioRad).

**Parasites**. *Leishmania mexicana* (MNYC/BZ/62/M379) derived strains were grown at 25 °C in HOMEM (Gibco) supplemented with 10% (v/v) heat-inactivated foetal calf serum (HIFCS) (Gibco) and 1% (v/v) Penicillin/Streptomycin solution (Sigma-Aldrich). Where required parasites were grown with selective antibiotics at the following concentrations: G418 (Neomycin) at 50 μgml−1; Hygromycin at 50 μg ml−1; Blasticidin S at 10 μg ml−1; Puromycin at 30 μg ml−1 (antibiotics from InvivoGen).

**CRISPR/Cas9**. Initial screening for bromodomain gene essentiality was performed with a modification of the approach developed by the Gluenz lab[32]. Per gene a single sgRNA was designed with EuPaGDT[66] to target the interior of the coding DNA sequence. Oligonucleotides are defined in Supplementary Data 1. Thirty residue homology flanks were identified adjacent to the CDS and appended to oligonucleotides designed to amplify drug resistance markers from blasticidin and neomycin drug resistance plasmids pGL2208 and pGL2663 respectively. After amplification of the sgRNA and resistance marker the PCR mixes were pooled and precipitated using standard ethanol precipitation, resuspended in sterile water and added to a transfection mix with $1 \times 10^7$ mid-log promastigotes. The cell line used was *L. mexicana T7/Cas9::HYG::SAT*[32]. Transfection was performed with an Amaxa Nucleofector 4D using program FI-115 and the Unstimulated Human T-Cell Kit. The mix was resuspended in 10 ml HOMEM 20% FCS and immediately split into two 5 ml aliquots. Following 6–18 h of recovery time the parasites were plated at 1:5, 1:50 and 1:500 dilutions in media containing the selective drug blasticidin or G418. Endogenous tagging was performed using the pPLOT 3xMyc::mNG BSD donor vector to install N-terminal tags to BDF5, preserving the 3′ UTR for native mRNA regulation (Oligonucleotides defined in Supplementary Data 1).

**DiCre**. DiCre strains for BDF5 were generated as previously described[36]. Briefly, the BDF5 CDS and flanking regions were assembled into floxing or knockout plasmids using Gateway cloning, BDF5 was cloned into pGL2314 to fuse a 6xHA C-terminal tag and flank with loxP sites (Oligonucleotides defined in Supplementary Data 1). The BDF5::6xHAflx was first integrated into parasites with clones being assessed for correct integration, correct genome copy number and inducibility of the excision of BDF5flx gene prior to the second round of transfections to delete the remaining wild-type allele. The same quality controls were performed when selecting final clones with the genotype *Leishmania mexicana DiCre::Puro Δbdf5::HYG::BDF5::6xHA::flx::BSD*.

Inducible deletion of BDF5flx in DiCre cell lines was initiated by the addition of 300 nM rapamycin (Abcam) to promastigotes cultures at $2 \times 10^5$ cells ml−1. Cells were grown for 48 h then passaged into new media at a concentration of $2 \times 10^5$ cells ml−1; induction was maintained by the addition of 100 nM of rapamycin to suppress escape mutants.

**Clonogenic assays**. For clonogenic assays, mid-log cells were counted and then diluted to 1 cell per 800 μl and plated out into 200 μl volumes in 3 × 96-well plates to yield approximately 100 clones. Cells were plated in media ± 100 nM rapamycin and incubated at 25 °C for 3 w before counting of viable colonies by both visual screening and microscopic analysis

**Addback strains**. To generate episomal addbacks the BDF5 CDS was amplified from *L. mexicana* genomic DNA and cloned into the pNUS C-Ter GFP NEO (pGL1132) using HiFi Assembly (NEB) to generate a complementation vector. This was used as a base for site-directed mutagenesis using the Q5 Mutagenesis product (NEB) to generate mutations in the conserved asparagine residues N90 (OL9577 and OL10352), N257 (OL9579 and OL10353) to phenylalanine in BDF5 BD5.1 and BD5.2 (Supplementary Data 1). Log-phase promastigotes were transfected with 2–5 µg plasmid DNA and maintained as population under G418 selection.

**Inducible overexpression**. An adaptation of a published method was performed whereby BDF5::GFP alleles generated for episomal addback were amplified using PCR primers OL11307 and OL11308, where the oligonucleotide included a directional loxP site as well as a homology region for HiFi Assembly into pRIB Neo (pGL1132). The vector backbone was linearised with PacI/PmeI double digest, separated by agarose gel electrophoresis and purified using QiaEx II gel extraction resin (Qiagen). Log-phase promastigotes were transfected with 1–5 µg and cloned by limiting dilution. Clones were induced to express BDF5::GFP or BDF5$^{NN>FF}$::GFP.

**Mouse infections**. All experiments were conducted according to the Animals (Scientific Procedures) Act of 1986, United Kingdom, and had approval from the University of York Animal Welfare and Ethical Review Body (AWERB) committee. All animal studies were ethically reviewed and carried out in accordance with Animals (Scientific Procedures) Act 1986 and the GSK Policy on the Care, Welfare and Treatment of Animals. Mid-log parasites were treated with 500 nM rapamycin and allowed to progress to stationary phase. These cultures were used to infect BALB/C mice at a dose of $2 \times 10^5$ parasites per footpad. Infections were allowed to progress for 8 w, at which point mice were euthanised and footpads and popliteal lymph nodes were dissected for mechanical disruption and determination of parasite burden by limiting dilution of footpad material into serial, two-fold dilutions in a 96-well plate, as previously described[67].

**Live-cell microscopy**. To image mNeonGreen::BDF5 $10^6$ mid-log cells were incubated with 1 µg ml$^{-1}$Hoechst 3342 for 20 min at 25 °C to stain DNA, harvested by centrifugation at $1200 \times g$ for 10 min and washed twice with PBS. Cell pellets were resuspended in 40 µl CyGel (BioStatus) and 10 µl settled onto SuperFrost+ Slides (Thermo) then cover slip applied. Cells were imaged using a Zeiss AxioObserver Inverted Microscope equipped with Colibri 7 narrow-band LED system and white LED for epifluorescent and white light imaging. Cells were imaged using the x63 or x100 oil immersion DIC II Plan Apochromat objectives. Hoechst signal was imaged using the 385 nm LED and filter set 49, mNeonGreen with the 469 nm LED and filter set 38. Z-stacks were obtained using the Zen Blue software to control the system and exported as.CZI files to be processed in ImageJ (FIJI) using the Microvolution blind deconvolution module. Wavelength parameters were set for Hoechst (497 nm) and mNeonGreen (517 nm) emission and refractive index parameters were defined for Cygel (1.37). Blind deconvolution was iterated 100 times using the scalar setting. Maximum intensity projections were then exported as were individual Z-planes for subpanels in TIFF format. Amastigotes in murine bone marrow derived macrophages grown on glass bottomed 35 mm dishes (Thermo Scientific) and imaged in FluoroBrite DMEM (Gibco) using a heated plate holder to maintain the samples at 35 °C. In this instance due to the short imaging duration $CO_2$ supplementation was not provided.

**XL-BioID**. BirA*::BDF5 and CLK2::BirA* spatial control parasites were grown to $4 \times 10^6$ ml$^{-1}$ at which point biotinylation was initiated in 3 replicates of each line, by adding biotin to 150 µM for 18 h. To perform BDF5-proximity biotinylation during the cell cycle, BDF5 was endogenously C-terminally tagged with miniTurbo at both alleles. Parasites were grown to early log-phase (~$2 \times 10^6$/ml) and synchronised with 0.4 mg/ml hydroxyurea for 18 h. Parasites were then washed twice in pre-warmed culture medium and resuspended at $4 \times 10^6$ ml$^{-1}$ to initiate synchronous progression through the cell cycle. Biotinylation time points were taken at 0, 4 and 8 h after hydroxyurea wash off. At each time point, biotin was added to 0.5 mM for 30 min biotinylation.

After biotinylation, parasites were washed twice in PBS and resuspended to a density of $4 \times 10^7$ ml$^{-1}$ in pre-warmed PBS. DSP cross-linker was added to 1 mM and in vivo crosslinking proceeded for 10 min at 25 °C. Cross-linking was quenched for 5 min by addition of Tris-HCl pH7.5 to a concentration of 20 mM. Parasites were harvested by centrifugation and pellets stored at −80 °C until lysis. A pellet of $4 \times 10^8$ parasites was used for each affinity purification which was lysed in 500 µl ice cold RIPA buffer containing 0.1 mM PMSF, 1 µg ml-1 pepstatin A, 1 µM E64, 0.4 mM 1–10 phenanthroline. In addition, every 10 ml of RIPA (0.1% sodium dodecyl sulfate, 0.5% sodium deoxycholate, 1% IgePal-CA-630, 0.1 mM EDTA, 125 mM NaCl, 50 mM Tris pH7.5) was supplemented with 200 µl proteoloc protease inhibitor cocktail containing w/v 2.16% 4-(2-aminoethyl)benzenesulfonyl fluoride hydrochloride, 0.047% aprotinin, 0.156% bestatin, 0.049% E-64, 0.084% Leupeptin, 0.093% Pepstatin A (Abcam), 3 tablets complete protease inhibitor EDTA free (Roche) and 1 tablet PhosSTOP (Roche). Lysates were sonicated with a microtip sonicator on ice for 3 rounds of 10 s each at an amplitude of 30. 1 µl of Benzonase (250 Units, Abcam) was added to each lysate, digestion of nucleic acids

proceeded for 10 min at RT followed by 50 min on ice. Lysates were clarified by centrifugation at 10,000 g for 10 min at 4 °C. For enrichment of biotinylated material, 100 µl of magnetic streptavidin bead suspension (1 mg of beads, Resyn Bioscience) was used for each affinity purification from $4 \times 10^8$ parasites. Biotinylated material was affinity purified by end-over-end rotation at 4 °C overnight. Beads were washed in 500 µl of the following for 5 min each: RIPA for 4 washes, 4 M urea in 50 mM triethyl ammonium bicarbonate (TEAB) pH8.5, 6 M urea in 50 mM TEAB pH8.5, 1 M KCl, 50 mM TEAB pH8.5. Beads from each affinity purification were then resuspended in 200 µl 50 mM TEAB pH8.5 containing 0.01% ProteaseMAX (Promega), 10 mM TCEP, 10 mM Iodoacetamide, 1 mM CaCl$_2$ and 500 ng Trypsin Lys-C (Promega). On bead digest was carried out overnight at 37 °C while shaking at 200 rpm. Supernatant from digests was retained and beads washed for 5 min in 50 µl water which was then added to the supernatant. Digests were acidified with trifluoroacetic acid (TFA) to a final concentration of 0.5% before centrifugation for 10 min at 17,000 g. Supernatant was desalted using in house prepared C18 desalting tips, elution volume was 60 µl. Desalted peptides were either dried for MS analysis (BirA experiments) or used for enrichment of proximal phosphopeptides (miniTurboID experiments).

**Proximal phosphopeptide enrichment**. Desalted peptides were pipette mixed to ensure a homogenous sample and 40% was removed and dried down for the 'Total' proximal protein sample. To the remaining 36 µl was added 51.2 µl acetonitrile, 10 µl 1 M glycolic acid and 5 µl TFA. Magnetic Ti-IMAC-HP beads (ReSyn Biosciences) were washed three times for 1 min in a 2x volume of loading buffer (0.1 M glycolic acid, 80% acetonitrile-ACN, 5% TFA). 10 µl of bead suspension (200 µg beads) were used to enrich phosphopeptides from each affinity purification. Peptides were added to beads and incubated with shaking at 800 rpm for 40 min at RT. Beads were washed for 2 min at 800 rpm in 100 µl loading buffer, then 100 µl 80% ACN 1% TFA and finally 100 µl 10% ACN 0.2% TFA. Phosphopeptides were eluted in $2 \times 40$ µl 1% NH4OH for 10 min shaking at 800 rpm. 2 µl TFA was added and peptides dried down for MS analysis.

**Mass spectrometry data acquisition**. For XL-BioID analysis peptides were loaded onto an UltiMate 3000 RSLCnano HPLC system (Thermo) equipped with a PepMap 100 Å C18, 5 µm trap column (300 µm × 5 mm, Thermo) and a PepMap, 2 µm, 100 Å, C18 EasyNano nanocapillary column (75 mm × 500 mm, Thermo). Separation used gradient elution of two solvents: solvent A, aqueous 1% (v:v) formic acid; solvent B, aqueous 80% (v:v) acetonitrile containing 1% (v:v) formic acid. The linear multi-step gradient profile was: 3–10% B over 8 min, 10–35% B over 115 min, 35–99% B over 30 min and then proceeded to wash with 99% solvent B for 4 min. For analysis of samples prepared to measure dynamics of proximal proteins and phosphosites during kinetochore assembly, peptides were loaded onto an mClass nanoflow UPLC system (Waters) equipped with a nanoEaze M/Z Symmetry 100 Å C 18, 5 µm trap column (180 µm × 20 mm, Waters) and a PepMap, 2 µm, 100 Å, C 18 EasyNano nanocapillary column (75 mm × 500 mm, Thermo). Separation used gradient elution of two solvents: solvent A, aqueous 0.1% (v:v) formic acid; solvent B, acetonitrile containing 0.1% (v:v) formic acid. The linear multi-step gradient profile for 'Total' protein was: 3–10% B over 8 min, 10–35% B over 115 min, 35–99% B over 30 min and then proceeded to wash with 99% solvent B for 4 min. For phosphopeptide enriched samples the following gradient profile was used: 3–10% B over 7 min, 10–35% B over 30 min, 35–99% B over 5 min and then proceeded to wash with 99% solvent B for 4 min. In all cases the trap wash solvent was aqueous 0.05% (v:v) trifluoroacetic acid and the trapping flow rate was 15 µL/min. The trap was washed for 5 min before switching flow to the capillary column. The flow rate for the capillary column was 300 nL/min and the column temperature was 40 °C. The column was returned to initial conditions and re-equilibrated for 15 min before subsequent injections.

The nanoLC system was interfaced with an Orbitrap Fusion hybrid mass spectrometer (Thermo) with an EasyNano ionisation source (Thermo). Positive ESI-MS and MS2 spectra were acquired using Xcalibur software (version 4.0, Thermo). Instrument source settings were: ion spray voltage, 1,900 V; sweep gas, 0 Arb; ion transfer tube temperature; 275 °C. MS1 spectra were acquired in the Orbitrap with: 120,000 resolution, scan range: m/z 375–1,500; AGC target, 4e5; max fill time, 100 ms. Data dependent acquisition was performed in top speed mode using a fixed 1 s cycle, selecting the most intense precursors with charge states 2–5. Easy-IC was used for internal calibration. Dynamic exclusion was performed for 50 s post precursor selection and a minimum threshold for fragmentation was set at 5e3. MS2 spectra were acquired in the linear ion trap with: scan rate, turbo; quadrupole isolation, 1.6 m/z; activation type, HCD; activation energy: 32%; AGC target, 5e3; first mass, 110 m/z; max fill time, 100 ms. Acquisitions were arranged by Xcalibur to inject ions for all available parallelizable time.

**Mass spectrometry data analysis**. Peak lists in.raw format were imported into Progenesis QI (Version 2.2., Waters) for peak picking and chromatographic alignment. A concatenated product ion peak list was exported in.mgf format for database searching against the *Leishmania mexicana* subset of the TriTrypDB (8,250 sequences; 5,180,224 residues) database, appended with common proteomic contaminants. Mascot Daemon (version 2.6.1, Matrix Science) was used to submit

searches to a locally-running copy of the Mascot program (Matrix Science Ltd., version 2.7.0.1). Search criteria specified: Enzyme, trypsin; Max missed cleavages, 2; Fixed modifications, Carbamidomethyl (C); Variable modifications, Oxidation (M), Phospho (STY), Acetyl (Protein N-term, K), Biotin (Protein N-term, K); Peptide tolerance, 3 ppm (# 13 C = 1); MS/MS tolerance, 0.5 Da; Instrument, ESI-TRAP. Peptide identifications were passed through the percolator algorithm to achieve a 1% false-discovery rate as assessed empirically by reverse database search, and individual matches filtered to require minimum expect scores of 0.05. The Mascot.XML results file was imported into Progenesis QI, and peptide identifications associated with precursor peak areas were mapped between acquisitions. Relative protein abundances were calculated using precursor ion areas from non-conflicting unique peptides. For total proteome data, only non-modified peptides were used for protein-level quantification. Statistical testing was performed in Progenesis QI, with the null hypothesis being peptides are of equal abundance among all samples. ANOVA-derived p-values from QI were converted to multiple test-corrected q-values within QI. For phosphopeptide identifications, Mascot-derived site localisation probabilities were used. To associate quantification data with site localisation probabilities, Mascot search results including raw spectral details and QI quantification data were exported separately in.csv format. Empty rows in the Mascot.csv were removed in R (3.6.1) before joining the data from the two.csv files using KNIME Analytics Platform (4.3.1 KNIME AG). The combined.csv was stripped of non-quantified peptides before calculating Hochberg and Benjamini FDR q-values from the original QI ANOVA p-values.

To determine BDF5-proximal proteins from the BirA* experiments, normalised protein label free peak areas were analysed with SAINTq[68] using the following parameters: normalize_control = FALSE, compress_n_ctrl = 100, compress_n_rep = 100. CLK2-BirA* parasites were used as the spatial reference, SAINTq was used to determine significantly enriched proteins in BirA*-BDF5 parasites vs. CLK2-BirA* parasites. False-discovery rate for identified proximal proteins was 1%.

For miniTurboID total proximal data, peptide ion quantification data was exported from Progenesis LFQ and missing value imputation was performed for each sample group, drawing values from a left shifted normal log2 intensity distribution to model low abundance proteins (mean = 12.1, sd = 1.5). For each protein, a mean intensity profile across all samples was calculated from peptide ion intensities. For each peptide ion, a pearson correlation coefficient was calculated between the mean protein intensity profile and the individual peptide ion intensity profile. Peptide ions with a correlation coefficient >0.4 were then summed to calculate the label free intensity of the parent protein. Protein intensities were log2 transformed and proximal proteins were determined with the limma package[69], comparing against KKT3-miniTurbo and using options trend = TRUE and robust = TRUE for the eBayes function. Multiple testing correction was carried out according to Benjamini & Hochberg, false-discovery rate for identified proximals was 1%.

For miniTurboID phosphosite proximal data, label free intensities were exported from Progenesis LFQ and missing values imputed by drawing values from a left shifted normal log2 intensity distribution to model low abundance phosphopeptides (mean = 4, sd = 1.2). Phosphosites were aggregated by summing intensities which were then log2 transformed. Proximal phosphosites were determined with the limma package comparing against KKT3-miniTurbo and using options trend = TRUE and robust = TRUE for the eBayes function. Multiple testing correction was carried out according to Benjamini & Hochberg, false-discovery rate for proximal phosphosites was 5%.

Hypothetical proteins were screened for remote structural homology using HHPRED[21] and Phyre2[70] to identify putative domains in these proteins.

**Co-immunoprecipitation**. Co-immunoprecipitation to confirm XL-BioID hits was performed by tagging candidate proteins with 3xHA::mCherry PURO using an adapted pPLOT vector (gifted by Ewan Parry, Walrad Lab.) in the *L. mexicana* T7/Cas9 3xMyc::mNG::BDF5 strain. Correct integration of the tag was confirmed by western blotting for the HA epitope. For pulldowns, 30 ml of mid-log cultures (~$1.5 \times 10^8$ cells) were harvested, by centrifugation at $1200 \times g$ for 10 min and resuspended in PBS. DSP reversible cross-linker (dithiobis(succinimidyl propionate) (Thermo) was added to 1 mM and incubated for 10 min at 26 ºC. Cross-linking was quenched by the addition of Tris pH7.5 to 20 mM and parasites washed with PBS. The cells were then lysed using 1x RIPA buffer (Thermo) supplemented with 3x HALT Protease inhibitors (Thermo) and $1 \times$ PhosSTOP (Roche), 2 µl (500 Units) BaseMuncher Endonuclease (Abcam). The lysate was sonicated $3 \times 10$ s at 40% amplitude using a probe sonicator (Sonics Vibra-Cell) and then clarified by centrifugation $10, 000 \times g$ for 10 min at 4 ºC. HA-tagged bait proteins were then immunoprecipitated by the addition of 30 µl anti-HA magnetic beads (Pierce) incubated for 2 h with rotation at 4 ºC. The beads were then washed 3 times using the supplemented RIPA lysis buffer and proteins eluted from the beads using 40 µl 1x LDS buffer supplemented with 250 mM DTT and heating to 60 ºC for 10 min. The eluted fractions were analysed for the presence of the BDF5 prey protein and intended bait proteins by western blotting for the Myc and HA epitopes respectively. Non-crosslinking immunoprecipitations were conducted with $3 \times 10^8$ promastigotes lysed in 400 µl 150 mM KCl with 0.2 % IGEPAL CA-630 (Sigma-Aldrich) and 3x HALT Protease inhibitors (Thermo). Samples were subjected to 3 cycles of sonication (10 s on, 30 s off) using a Bioruptor Pico (Diagenode). Insoluble material was removed by centrifugation at $10, 000 \times g$ for 10 min at 4 ºC.

A-tagged bait proteins were then immunoprecipitated by the addition of 30 µl anti-HA magnetic beads (Pierce) incubated for 1 h with rotation at 4 ºC. Beads were then washed $3 \times 5$ min with lysis buffer, interacting proteins were eluted and analysed as per crosslinking immunoprecipitations.

**ChIP-Seq**. BDF5 ChIP-seq was performed using a modification of a protocol optimised for *T. brucei*[50] and the ChIP-it Express Enzymatic Kit (Active Motif). *Lmx DiCre* or *Lmx DiCre BDF5::6xHA*$^{-/+flx}$ parasites were grown to $5 \times 10^6$ cells ml$^{-1}$ in sufficient volume to collect $3 \times 10^8$ cells per ChIP replicate. Cells were fixed with 1% formaldehyde for 5 min then quenched with 1x of the included glycine solution. Fixed cells were Dounce homogenised until only nuclei were visible by microscopy. Following enzymatic digestion of purified nuclei, the mix was sonicated $3 \times 10$ s at 40% amplitude using a probe sonicator (Sonics Vibra-Cell) to increase recovery of mono- to tetra-nucleosomes[44]. Chromatin fractionation and release was checked by agarose gel electrophoresis before immunoprecipitation using commercially conjugated Pierce anti-HA magnetic beads (Thermo, 88836) for 2 h at 4 ºC. The antibody conjugated to the beads is a mouse monoclonal IgG1, clone 2-2.2.14, 30 µl beads were used per IP. Beads were washed and the crosslinking was reversed following the manufacturer's instructions. Liberated DNA and the retained input samples were purified and concentrated using ChIP-cleanup mini-columns (Zymogen). This DNA was quantified using a Qubit (Qiagen) High Sensitivity DNA kit and sent for library preparation. Library generation was performed on a minimum of 5 ng DNA using NEBNext DNA Ultra II library prep kit for Illumina, with dual 8 bp indexing in the Genomics Laboratory of University of York Bioscience Technology Facility. Sequencing was conducted at the University of Leeds. Reads were quality checked and trimmed using FastQC version 11.0.5 and Cutadapt version 2.5, respectively. This was followed by alignment to the *L. mexicana* MHOMGT2001U1103 Version 44 genome using BWA-MEM (version 0.7.17). BWA MEM was run with default parameters, including a minimum seed length of 19 bp and no filter on multi-mapping reads. Paired ChIP-seq and input alignment files were normalised to each other using deepTools' bamCompare (version 3.3.1) with SES normalisation and bin size of 500. Bigwig files were converted to wig files with UCSC's bigWigToWig tool, and the resulting 3 files were combined by taking the mean. Peaks were filtered to only include those with a mean log2 ratio greater than 0.5 and peaks that were less than 5 kb apart were merged. Strand switch regions were defined as regions between the end of a CDS on one strand and the beginning of CDS on the other strand. Data were visualised using IGV (Broad Institute) and Circa software (OMGenomics).

**Flow cytometry**. Flow cytometry of fixed and live cells treated with propidium iodide for cell cycle and live/dead analysis was conducted as previously reported[36]. For determination of total RNA levels by SYTO RNASelect staining, 1 ml of culture was treated with 500 nM SYTO RNASelect for 20 min at 25 ºC. Cells were collected by centrifugation $1200 \times g$ for 10 min and washed with PBS before resuspension in PBS 10 mM EDTA pH 7.4. Cells were analysed using a Beckman Coulter Cyan ADP flow cytometer with detection of the stained RNA in the FL1 channel. Data was analysed using FCS Express 7 Flow version 7.10.0007 (De Novo Software, Inc.) using the Multicycle DNA module (model 6) for cell-cycle analysis.

**Stranded ERCC controlled RNA-seq**. Cultures of promastigote *Lmx DiCre BDF5::6xHA*$^{-/+flx}$ were treated with DMSO or 300 nM rapamycin for 48 h then passaged to a density of $2 \times 10^5$ cells ml$^{-1}$ for another 48 h. At this point $2 \times 10^7$ cells were collected, washed in PBS and processed for total RNA extraction. Total RNA was extracted using Monarch Total RNA Miniprep kit (NEB) as per the manufacturer's instructions with the exception of the addition of ERCC Synthetic RNA Transcripts (Ambion) to the RNA extraction buffer used to lyse the cells. This was added to a final concentration of 1/1000 from the manufacturers stock solution. In addition to the on-column Dnase digest, an additional treatment of the eluted RNA was performed with TURBO DNA Free kit as per the manufacturer's protocol. RNA was processed by Novogene using Illumina Ribo Zero method and NEBNext® Ultra™ Directional RNA Library Prep Kit to generate libraries which were sequenced on Illumina NovaSeq 6000 S4 flowcell with PE150. Reads were processed with FASTQC Groomer before mapping with HiSAT. BAM files were converted to bigwig format using bamCoverage (DeepTools) with a scaling factor applied to normalise the total reads to the median ERCC read values. Metaplots were generated using deepTools computematrix and plotProfile tools for dSSR and cSSR in reference-point mode (centre point of the SSR) using 500 bp bins. PTU metaplots were generated using the same tools in scale-regions mode.

**Spike-In controlled RT-qPCR**. Instead of ERCC RNAs, yeast RNA was used to spike-in control RT-qPCR assays. Yeast total RNA was extracted from a haploid wild-type *Saccharomyces cerevisiae* SEY6210, kindly gifted by Dr Chris MacDonald, University of York. Cultures of promastigote *Lmx DiCre BDF5::6xHA*$^{-/+flx}$ were treated with DMSO or 300 nM rapamycin for 48 h then passaged to a density of $2 \times 10^5$ cells ml$^{-1}$ for another 48 h. At 96 h, $2 \times 10^7$ cells were collected, washed in PBS and processed for total RNA extraction. Total RNA was extracted using Monarch Total RNA Miniprep kit (NEB) as per the manufacturer's instructions with the exception of the addition of 200 ng of Yeast total RNA per 300 µl of lysis buffer. Following total RNA extraction, gDNA was removed using TURBO DNA-

free treatment (Invitrogen) standard protocol. RT-qPCR reactions wer set up using Luna Universal One-Step RT-qPCR Kit (NEB), using 100 nanograms of total RNA as template. Reactions were amplified and measured using the SYBR and ROX channels of an Applied Biosystems QuantStudio 3 System machine. Oligonucleotides were designed using Primer-BLAST against the 18s rRNA, Cyclophilin A, and tRNA$^{Lys}$ genes (Supplementary Data 1), primers against Yeast ACT1 were used to quantify the exogenous control[71]. Primer efficiencies were verified to be between 95% and 105% using a standard curve analysis prior to relative quantitation experiments, melt curves were performed to ensure amplification of single products. Samples for comparison were run in technical quintuplicates. No-reverse transcriptase and no-template controls were included on each plate for each sample condition, each in duplicate. Relative quantitation experiments were performed using the ΔΔct method in the RQ module of ThermoFisher Cloud including the $T$ test option for comparing induced to non-induced samples.

**Splicing RT-PCR.** Analog-sensitive CRK9 (LmxM.27.1940) mutants were generated using Cas9 to perform precise genome editing to replace the codon encoding methionine at the gatekeeper position (M501) with a glycine or alanine residue (protocol adapted from[60]). Oligos sequences provided in Supplementary Data 1. The validation of the CRK9 mutants was performed by sequencing using OL11605 and a dose response curve, set at $2.5 \times 10^4$ cells ml$^{-1}$ treated with the bulky kinase inhibitors (BKIs: PP1 (1-(1,1-dimethylethyl)−3-(4-methylphenyl)−1H-pyr-azolo[3,4-d]pyrimidin-4-amine), 1NM-PP1 (1-(1,1-dimethylethyl)−3-(1-naphtha-lenylmethyl)−1H-pyrazolo[3,4-d]pyrimidin-4-amine) and 1NA-PP1 (1-(1,1-dimethylethyl)−3-(1-naphthalenyl)−1H-pyrazolo[3,4-d]pyrimidin-4-amine)) in a range concentration varying from 0 to 120 µM. The viability of treated and untreated control was assessed after 72 h, using Alamar blue at 0.0025% (w/v). The parental T7/Cas9 cell line was used as control for the analog-sensitive CRK9 lines. The inhibition profile was analysed by nonlinear regression using Prism Version 9 (GraphPad).

Cultures of promastigote *Lmx DiCre BDF5::6xHA*$^{-/+flx}$ were treated with DMSO or 300 nM rapamycin for 48 h then passaged to a density of $2 \times 10^5$ cells ml$^{-1}$ for another 48 h. Positive controls for cis- and trans- splicing defects were provided by treating *L. mexicana T7/Cas9* CRK9$^{M501G}$ with 30 µM 1NM-PP1 for 3 h (15x the EC90 at 72 h). Total RNA was purified using NEB Monarch Total RNA MiniPrep Kit. cDNA was synthesised using NEB ProtoScript II with random hexamers. Triple-primer PCR was conducted to determine the trans-splicing of the SL RNA to LmxM.25.0910 with OL12370, OL12371 and OL12372. Cis-splicing of the intron in polyA-polymerase (LmxM.08_29.2600) was detected using OL OL12342 and OL12343. PCRs were performed with PCRBio Ultra Red Mix.

**Statistics.** For routine statistical analyses data were analysed with Prism Version 9 (GraphPad). Western blot quantitation was performed using BioRad Imagelab software 6.0 using the Stain-Free Total Protein Channel as the normalisation channel.

**Reporting summary.** Further information on research design is available in the Nature Research Reporting Summary linked to this article.

## Data availability

The data that support this study are available from the corresponding author upon reasonable request. The mass spectrometry data sets and proteomic identifications are available to download from MassIVE (MSV000087750 [https://massive.ucsd.edu/ProteoSAFe/dataset.jsp?task=60fc23becb8a4dad8d954ec33d0ce5ff]) and ProteomeXchange (PXD027080). The ChIP-Seq and RNA-Seq reads are available as FASTQ files at the European Nucleotide Archive under the accession code PRJEB46800. Structures of Leishmania donovani BDF5 bromodomains were retrieved from the Protein Data Bank (https://www.rcsb.org/) using the PDB IDs 5TCM and 5TCK. Genome FASTA files and GFF files were retrieved from TriTrypDB [https://tritrypdb.org/tritrypdb/app]. Source data are provided with this paper.

## Code availability

Custom Python code used to identify the BDF5 ChIP-seq peaks is provided with this paper as a Supplementary Software text document.

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

## Acknowledgements

This work was supported by funding from GSK through the Pipeline Futures Group and a Fellowship from a Research Council United Kingdom Grand Challenges Research Funder under grant agreement 'A Global Network for Neglected Tropical Diseases' grant number MR/P027989/1. to Nathaniel Jones. This work was part-funded by the Wellcome Trust [ref: 204829] through the Centre for Future Health (CFH) at the University of York. Elmarie Myburgh, University of York, assisted with lymph-node dissection and parasite burden assays. We thank our colleagues in The Bioscience Technology Facility of the University of York, who provided expertise and technical support that assisted this work, and Robert Kirkpatrick (former GSK employee) for his critical input to develop the collaboration between University of York and GSK.

## Author contributions

N.G.J., F.C., A.J.W. and J.C.M. conceived the project. J.C.M., A.J.W., R.G., J.M. and F.C. supervised the project. N.G.J. and J.C.M. designed the experiments. N.G.J., V.G., G.M. and J.B.T.C. performed the experiments. N.G.J., V.G., G.M. and K.N. analysed experimental data. N.G.J. wrote the manuscript and all other authors revised it. N.G.J., R.P., I.R., F.C., A.J.W. and J.C.M. acquired funding.

## Competing interests

This study had funding support and intellectual input from collaborators at GSK. We disclose that Félix Calderón, Raquel Gabarró, Julio Martín, Rab Prinjha, Inmaculada Rioja are employees of GSK. We declare no other conflicts of interest.
