## [Peer Review File · Nature Communications]

REVIEWER COMMENTS

Reviewer #1 (Remarks to the Author):

In the manuscript entitled “Bromodomain factor 5 is an essential transcriptional regulator of the Leishmania genome” Jones et al. have initiated the first studies of the bromodomain factors of Leishmania species. They identify seven BDFs by genome sequence annotation, and using the approach of creating knockout lines they demonstrate five of these (BDFs 1-5) to be essential. Thereafter the study pursues one of these, BDF5, in determining its cellular role. Using ChIP-Seq studies the authors find Transcriptional Start Regions (TSRs) to be enriched in BDF5, and RNA-seq analyses of BDF5-depleted cells reveal a global decrease in PolII-transcript levels. A study of the BDF5-proximal proteome suggests that BDF5 may be involved in other processes as well, though the authors find no role in the DNA damage response or in RNA splicing events.

The work is important to the parasitology field, particularly the field of Leishmania biology, where transcriptional events still remain poorly understood and no work about the bromodomain factors has been published. The data are technically sound, with appropriate controls included in all experiments, and the conclusions drawn are in consonance with the presented data.

Comments:

1) What about the possible effects of BDF5 deletion on PolII- and PolIII-mediated transcription? Real time PCR analyses could be carried out to check this for some of the genes.

2) Regarding data in Figs. S6 and 2E:

The data in 2E indicates significantly decreased parasite burden from footpads infected with BDF5-depleted cells as compared to wild type cells at the end of 8 weeks; these parasites are likely the ones which survived as dropout did not occur in some cells, reflected in the PCRs in S6E.

S6A shows dropout has occurred and no hint of any non-dropouts are detected in the PCRs, therefore in the vast majority of these stationary phase cells (lines 294-295) the BDF5 gene is now absent, and these are now injected into mouse footpad. Even if the dropout did not occur in some cells, as it was not detectable in the PCRs in S6A, the effect of the BDF5 dropout should be visible in S6B when compared to wild type.

What is the data in Fig S6B, it looks like it is the footpad thickness/size and not the footpad lesion size as at 0 weeks itself it is at 2 mm. This is also what the y axis says, and if so, must be corrected in the figure legend as well as on lines 296-297.

Looking at the change in footpad thickness/size in animals infected with wild type parasites over 8 weeks, there is hardly any increase. The low number of injected parasites per footpad (2×10^5) may be the reason for this. Typically 5×10^6 or more parasites are injected. With hardly any change in footpad thickness/size over 8 weeks the infections with wild type parasites, it is difficult to estimate differences in infections with BDF5-depleted parasites. This experiment needs to be revisited, ensuring infections are set up only with metacyclics, and with 20-40 times more parasites.

Minor Comments:

- 1) In the co-IP experiments a cross-linking agent was used. Has any co-IP been done without the cross-linking agent; that would pick up the stablest interactions.
 - 2) There is no Figure 5 in the paper!!!
 - 3) Line 239 should read Fig.S3A, S3B.
 - 4) Fig. S3C and S3D and discussion of the data in them do not figure anywhere in the text.
 - 5) Line 604: should read H2AZ acetylation levels (HAT1 depletion has no effect on H2AZ deposition).
 - 6) Line 618: reference citations to be modified: ref. 15 is about HAT4; and ref. 69 carried out assays with crude Leishmania whole cell lysates (carrying all four HATs) with overexpressed HAT2, and with these extracts found acetylation at H4K4 in vitro. Ref.18 carried out the assay with HAT2 overexpressed in Leishmania and pulled down from lysates of these cells, and found it to acetylate H4K10 (subsequently shown to be the case in vivo also).
 - 7) Lines 619-620: transcription initiation positioning was not examined, only effects on levels of global transcription. Reduction of H4K10Ac levels at TSRs in HAT2 +/- cells did not affect global transcript levels. Of the few genes that were downregulated upon H4K10Ac depletion, CYC4 and CYC9 were found to be activated in cell cycle-dependent manner (CYC4 in S phase and CYC9 in G2/M).
 - 8) Fig. S1A: font size too small, not legible even at 150% magnification.
 - 9) Header for Fig. S3 legend: BD5 to read BDF5
 - 10) Fig. 6B legend: line 1008. Footpad lesion size should read footpad size?
 - 11) Fig. S8 legend: line 1037: BDDF5 to read BDF5
- Fig. S11 legend: line 1070: alarm to read Alamar

Reviewer #2 (Remarks to the Author):

The excellent work described in this paper focuses on chromatin interacting readers of acetylated lysines in the Leishmania parasite. Kinetoplastids, of which Leishmania is a member, diverged early from many well studied model organisms, and thus represent an opportunity to understand the evolution of gene regulatory mechanisms. In particular, gene regulation in kinetoplastids has a number of unusual features, such as polycistronic transcription units encompassing large numbers of genes with unrelated functions and a paucity of DNA sequence specific transcription factors as classically defined in other systems. Kinetoplastids do possess histone modifications, and the mechanisms for how these histone modifications influence gene regulation are still largely unknown. Bromodomain proteins, which possess domains that can interact with acetylated lysines, have been studied in *T. cruzi* and *T. brucei*, but so far they remain largely uncharacterized in Leishmania.

This paper makes very important contributions to understanding the gene regulatory role of bromodomain factor 5 (Bdf5) in Leishmania parasites. The authors use a truly impressive array of techniques to characterize Bdf5 genomic localization, Bdf5 binding partners, and the functional role for Bdf5 in promoting polIII transcription using inducible genetic ablation. In addition, the authors examine the implications for virulence in the absence of Bdf5 using a mouse model of infection, and find that virulence is significantly decreased following Bdf5 ablation. They convincingly show that Bdf5 localizes to transcription start sites as seen in other kinetoplastids but has additional sites of localization elsewhere at transcription termination sites and at tRNA loci. They also characterize a set of conserved interacting partners of Bdf5, which they call Conserved Regulators of Kinetoplastid Transcription (CRKT) complex. Finally, they show that Bdf5 is essential for maintaining high levels of polIII transcripts throughout the genome, which provides an explanation for why Bdf5 ablation rapidly kills parasites.

What is particularly impressive is that after discovering a rather dizzying array of interacting partners for Bdf5, the authors (1) verified many of these interactions using co-IP and (2) went several steps further in understanding which of these interactions represent core functional features for Bdf5. That is to say, after discovering that Bdf5 can interact with a number of proteins implicated in different processes such as DNA repair, splicing, and transcription, the authors went on to show that DNA repair and splicing remain largely unaffected by Bdf5 deletion, whereas pol II driven transcript levels are nearly halved following ablation of the protein. This additional functional characterization really strengthens the paper, and avoids the problem of 'cataloguing' where lots of partners are presented and it's difficult to know which are the most important. The discoveries presented here represent an important first step in understanding the role for bromodomain proteins in Leishmania. Notably, while some features of bromodomain proteins are shared with *T. brucei* in that Bdf5 localizes to divergent strand switch regions that are thought to be sites of transcript initiation, there are other features that appear unique to Leishmania, including the lack of double peaks at these sites as well as the appearance of the protein at

some transcription termination sites, which has not been observed in *T. brucei*. In addition, while some Bdf5 complex members are shared between *Leishmania* and *T. brucei*, others are not.

The findings here will serve as an excellent jumping off point for the further characterization of other bromodomain proteins in *Leishmania*, while also making important contributions to the understanding of how gene regulatory mechanisms operate in this early branching organism. Thus, the paper is appropriate to the wide readership offered by *Nature Communications*.

In general, the experiments presented in the paper are technically sound and are very well controlled. I especially appreciated the add backs of the wildtype and mutant versions of Bdf5 and the use of the diCre system. I have only a few major suggestions and a number of smaller points that I'm hoping will not be too onerous in revision.

Major Points

I may have missed it, but I don't think the ChIP-seq experiments included a control in the form of a pulldown with a nonspecific (IgG) antibody or an HA pulldown in an untagged parasite line. It would be nice to see one or the other of these controls, though certainly both are not necessary. At least in my hands, some peaks are called by MACS when comparing the IP material to the input control, but these same peaks appear in the IgG or the HA control pulldowns. Particularly because Bdf5 showed up in some rather surprising regions, it would be good to verify that these peaks of localization are genuine and not some artifact of sticky DNA or something else. If the authors have performed lots of ChIP-seq using this same workflow and have such a control on hand that would be ideal. Otherwise it might make sense to perform at least one replicate of one of these controls to validate all the called peaks in the experimental samples. Bedtools can easily be used to filter out peaks that appear in the control IPs.

Figure 3. The visualization for Bdf5 localization in all the *Leishmania* chromosomes is neat, but there's so much going on that I think the main points about the localization for Bdf5 are not easy to see using this visualization. The authors might consider swapping in Figure S7 as the main figure in the paper while putting the current Figure 3 as the supplement.

The experiments presented in Figure S11 and S12 are really hard to understand without some description of the assay in the text. If length constraints are preventing the authors from including this in the main text, it would be great to include it as supplemental text somewhere. I think the result that Bdf5 ablation doesn't affect splicing is an important one, and is given rather short shrift as presented here.

Minor Points.

1. Something I found slightly surprising was that in the result for Figure 2E, the median levels of parasite burden are 10-fold lower in the DMSO treated Bdf5-/+flx pNUSBdf5 compared to the DMSO treated Bdf5+/-flx condition. I noticed the same thing for the result presented in Figure S6C. I think this difference was mentioned in the text but could the authors speculate a bit more on what's going on here? Is the difference statistically significant? Is there some leaky flipping of the pNUS allele (doesn't seem so from the gels) or is there some other explanation the authors could postulate?
2. Line 239, I think the figure callout should be S3A, S3B rather than S2A, S2B.
3. In the paragraph starting on line 264, the authors discuss that the Bdf5 add back doesn't fully complement. Could it be that the addback is missing some regulatory mechanism that exists for the endogenous version? Might it be possible to speculate on why the complementation was not complete? If not, that's ok, I realize some results are just puzzling and we occasionally have to accept that.
4. Line 92 should read transcriptional start sites rather than transcriptionally start sites.
5. In the section starting at line 324, can the authors be a little more explicit about the controls used to call peaks (input samples) and how peaks were specifically called (MACS?).
6. I think there may be a stray 'which' in line 387, or there is something weird going on with that sentence.
7. I'm so impressed with all the co-IPs conducted. Since so many produced nice results, would it be possible to include something in the text such as 'of the x interactions we tested were able to verify Y of them.'
8. In the co-IP S8 Figure, would it be possible to put the predicted size in parentheses next to the label for each protein tested? Since many co-IPs produced multiple bands on the gel for the bait protein, it's sometimes difficult to ascertain which band is the relevant one to be looking at. In addition, some of the bands for Bdf5 are rather faint. It would be helpful to indicate which interactions the authors deemed verified by including a star or a plus sample by the relevant lanes.

9. Figure 6B, The label for chromosome 8 has a typo.

10. In the ChIP-seq methods section, could just a little more detail be included for how these samples were analyzed? For example, how many mismatches were allowed during alignment, and were reads aligned uniquely or not? One figure legend mentions using MACS to call peaks but I don't see that represented here. If MACS was used what mode was run: broad? Narrow?

11. I can't find any mention of Figure S3 in the text, as noted again below in the Figure legend section. Are the G1, S, and G2 phases gated here?

12. In general, the figure legends could benefit with some revision so that they can stand alone without having to refer to the text. I think all the figured legends should be carefully looked through, but I'll cite some examples of below.

Examples for figure legends:

Figure 2A should include a description of how parasites were treated for 48 hours, then diluted, then treated for the remaining time, etc.

Figure 2C legend could be replaced with 'Western blot showing levels of BDF5::6XHA protein following treatment with either DMSO or Rapamycin for 72 hours. The 3 samples shown represent biological replicates.' (Are these biological replicates, I wasn't sure???)

Figure 2E should mention that parasites were pretreated with DMSO or Rapamycin and then used to infect mice, etc.

Figure S1A could use a better description of the cartoon. All the cartoons work really well and their inclusion is much appreciated!

Figure S2 legend should define TCM and TCK as PDB IDs.

Figure S3. Are these biological replicates? Are the results quantified somewhere? I can't find any reference to Figure S3 in the text, though maybe I missed it.

Figure S5. It wasn't clear to me if the pRIB construct was randomly integrated or targeted somewhere into the Lmx:DiCre line.

Figure S7. I'm not sure what the logEvsI track represents, please include in legend.

Figure S9. Missing legend for panel E.

Reviewer #3 (Remarks to the Author):

The manuscript by Jones et al. examines bromodomain factors (BDF) of Leishmania. They present thorough description into the identification of BDF containing proteins and whether they are required for viability. They go on to provide extensive functional characterization of BDF5. Overall, this is a really nice manuscript that is technically sound and I believe will be of broad interest.

Minor Comments:

- Figure 4 is overall a nice summary of the BioID results. One minor point is that the Dash Outline denoting Co-IP verified is hard to see.

- Supplemental datasets of BioID data would benefit from a brief legend describing the columns contained in each file.

- miniTurboID experiments are not mentioned in the methods.

- More complete methods for BioID including mass spectrometry details, number of replicates, and statistical analysis to determine enrichment of BDF5 over control would be nice.

- Likely nothing further is needed for this manuscript but the phosphoproteomic miniTurboID data sampled over time seems under explored.

Firstly, we thank the reviewers for taking the time to read and review our manuscript. We are grateful for the positive and constructive feedback, which highlighted several areas to develop the manuscript. We hope that our responses and changes to the manuscript have improved it in line with their suggestions.

REVIEWER COMMENTS

Reviewer #1 (Remarks to the Author):

Comments:

1) What about the possible effects of BDF5 deletion on PolI- and PolIII-mediated transcription? Real time PCR analyses could be carried out to check this for some of the genes.

This was a good suggestion and something we thought might answer the question of the breadth of BDF5's role in transcriptional regulation. We have conducted spike-in controlled qRT-PCR of 3 reporter genes (18s rRNA, Cyclophilin A, and tRNA^{Lys}) which revealed that only the Pol2 transcribed Cyclophilin A gene was significantly downregulated after BDF5 inducible deletion. This data is included as a new panel in Figure 6 and a method has been included.

2) Regarding data in Figs. S6 and 2E:

The data in 2E indicates significantly decreased parasite burden from footpads infected with BDF5-depleted cells as compared to wild type cells at the end of 8 weeks; these parasites are likely the ones which survived as dropout did not occur in some cells, reflected in the PCRs in S6E. S6A shows dropout has occurred and no hint of any non-dropouts are detected in the PCRs, therefore in the vast majority of these stationary phase cells (lines 294-295) the BDF5 gene is now absent, and these are now injected into mouse footpad. Even if the dropout did not occur in some cells, as it was not detectable in the PCRs in S6A, the effect of the BDF5 dropout should be visible in S6B when compared to wild type.

We have provided a contrast adjusted gel image in Supplemental Figure 6A to highlight that there still remains a band corresponding to an intact BDF5 locus in the inoculum. The end-point PCR is really only qualitative, as efficiency of amplification of the smaller fragment might come to dominate the reaction over the longer intact locus, and not giving a quantitative representation of the relative amounts of BDF5 replete or null individuals. Results from the clonogenic survival assays suggest that there is a subpopulation, probably 2-5% of cells, that do not excise the floxed BDF5 allele. These parasites likely go on to generate the infection in the mice infected with the Rapamycin-treated parasites. The DiCre system is unfortunately not perfect, but it remains the best inducible system available investigating essential Leishmania genes.

What is the data in Fig S6B, it looks like it is the footpad thickness/size and not the footpad lesion size as at 0 weeks itself it is at 2 mm. This is also what the y axis says, and if so, must be corrected in the figure legend as well as on lines 296-297.

We apologise for the confusion here and have adjusted the legend accordingly to "Footpad Size".

Looking at the change in footpad thickness/size in animals infected with wild type parasites over 8 weeks, there is hardly any increase. The low number of injected parasites per footpad (2x10⁵) may be the reason for this. Typically 5x10⁶ or more parasites are injected. With hardly any change in footpad thickness/size over 8 weeks the infections with wild type parasites, it is difficult to estimate differences in infections with BDF5-depleted parasites. This experiment needs to be revisited, ensuring infections are set up only with metacyclics, and with 20-40 times more parasites.

With Leishmania mexicana it is typically not required to purify metacyclic promastigotes (Damianou et al, PLoS Pathogens, 2020- PMID 32544189) prior to initiating murine infections, so in this instance we used stationary phase cultures. Although the footpad lesions were small, they were consistent in size with those reported by Damianou et al. using the same parental DiCre strain for a DUB2 inducible null. Despite the small lesion size, we were able to detect significant differences in the parasite burden between the groups infected with induced and non-induced parasites. Overall, as the existing data suggested BDF5 was important for infection and survival in mice, we were not inclined to expand animal work to investigate lesion pathology; so as to align with NC3Rs guidance to reduce the numbers of animals used in biological research.

Minor Comments:

1) In the co-IP experiments a cross-linking agent was used. Has any co-IP been done without the cross-linking agent; that would pick up the stablest interactions.

We agree this is important to fully understand the stability of the CRKT complex. We have performed non-crosslinking immune precipitations of 7 members of the CRKT complex, and the control strain KKT19 which validate these interactions as stable under native conditions, with the exception of BDF3. We cannot exclude that the epitope tags on the two proteins does not destabilise the BDF3-BDF5 interaction. KKT19 again did not co-

precipitate BDF5 highlighting its value as a control. This data has been added as a new panel to Supplemental Figure 7. A method for this approach has been added.

2) There is no Figure 5 in the paper!!!

This has been corrected.

3) Line 239 should read Fig.S3A, S3B.

Corrected

4) Fig. S3C and S3D and discussion of the data in them do not figure anywhere in the text.

We have added the following at line 260. "Deletion of BDF5 did not introduce a specific cell cycle defect, although induced cultures appeared to have a reduced number of G1 arrested cells at 72h post induction (Fig. S3C). The proportion of non-viable cells in the cultures at this point was ~10% (Fig. S3D) indicating and in combination with other experiments indicates that BDF5 depletion leads to a rapid cytostatic phenotype followed by eventual cell death."

5) Line 604: should read H2AZ acetylation levels (HAT1 depletion has no effect on H2AZ deposition).

We have added this correction.

6) Line 618: reference citations to be modified: ref. 15 is about HAT4; and ref. 69 carried out assays with crude Leishmania whole cell lysates (carrying all four HATs) with overexpressed HAT2, and with these extracts found acetylation at H4K4 in vitro. Ref.18 carried out the assay with HAT2 overexpressed in Leishmania and pulled down from lysates of these cells, and found it to acetylate H4K10 (subsequently shown to be the case in vivo also).

Corrected

7) Lines 619-620: transcription initiation positioning was not examined, only effects on levels of global transcription. Reduction of H4K10Ac levels at TSRs in HAT2 -/+ cells did not affect global transcript levels. Of the few genes that were downregulated upon H4K10Ac depletion, CYC4 and CYC9 were found to be activated in cell cycle-dependent manner (CYC4 in S phase and CYC9 in G2/M).

This section has been re-written for clarity.:

"Transcription initiation positioning at TSRs was not examined but the reduction of H4K10ac in HAT2-/+ background did not lead to global transcription reduction. However, expression of the Cyclin 4 and Cyclin 9 mRNA was reduced in this mutant. The regions upstream of these genes were found to be enriched for H4K10ac, and this was also reduced in the HAT2-/+ strain. Intriguingly, transcription of these genes was cell cycle dependent."

8) Fig. S1A: font size too small, not legible even at 150% magnification.

Font size has been increased.

9) Header for Fig. S3 legend: BD5 to read BDF5

Corrected.

10) Fig. 6B legend: line 1008. Footpad lesion size should read footpad size?

Corrected.

11) Fig. S8 legend: line 1037: BDDF5 to read BDF5

Corrected.

Fig. S11 legend: line 1070: alarm to read Alamar

Corrected.

Reviewer #2 (Remarks to the Author):

The excellent work described in this paper focuses on chromatin interacting readers of acetylated lysines in the Leishmania parasite. Kinetoplastids, of which Leishmania is a member, diverged early from many well studied model organisms, and thus represent an opportunity to understand the evolution of gene regulatory mechanisms. In particular, gene regulation in kinetoplastids has a number of unusual features, such as polycistronic transcription units encompassing large numbers of genes with unrelated functions and a paucity of DNA sequence specific transcription factors as classically defined in other systems. Kinetoplastids do possess histone modifications, and the mechanisms for how these histone modifications influence gene regulation are still largely unknown. Bromodomain proteins, which possess domains that can interact with acetylated lysines, have been studied in *T. cruzi* and *T. brucei*, but so far they remain largely uncharacterized in Leishmania.

This paper makes very important contributions to understanding the gene regulatory role of bromodomain factor 5 (Bdf5) in *Leishmania* parasites. The authors use a truly impressive array of techniques to characterize Bdf5 genomic localization, Bdf5 binding partners, and the functional role for Bdf5 in promoting polII transcription using inducible genetic ablation. In addition, the authors examine the implications for virulence in the absence of Bdf5 using a mouse model of infection, and find that virulence is significantly decreased following Bdf5 ablation. They convincingly show that Bdf5 localizes to transcription start sites as seen in other kinetoplastids but has additional sites of localization elsewhere at transcription termination sites and at tRNA loci. They also characterize a set of conserved interacting partners of Bdf5, which they call Conserved Regulators of Kinetoplastid Transcription (CRKT) complex. Finally, they show that Bdf5 is essential for maintaining high levels of polII transcripts throughout the genome, which provides an explanation for why Bdf5 ablation rapidly kills parasites.

What is particularly impressive is that after discovering a rather dizzying array of interacting partners for Bdf5, the authors (1) verified many of these interactions using co-IP and (2) went several steps further in understanding which of these interactions represent core functional features for Bdf5. That is to say, after discovering that Bdf5 can interact with a number of proteins implicated in different processes such as DNA repair, splicing, and transcription, the authors went on to show that DNA repair and splicing remain largely unaffected by Bdf5 deletion, whereas pol II driven transcript levels are nearly halved following ablation of the protein. This additional functional characterization really strengthens the paper, and avoids the problem of 'cataloguing' where lots of partners are presented and it's difficult to know which are the most important. The discoveries presented here represent an important first step in understanding the role for bromodomain proteins in *Leishmania*. Notably, while some features of bromodomain proteins are shared with *T. brucei* in that Bdf5 localizes to divergent strand switch regions that are thought to be sites of transcript initiation, there are other features that appear unique to *Leishmania*, including the lack of double peaks at these sites as well as the appearance of the protein at some transcription termination sites, which has not been observed in *T. brucei*. In addition, while some Bdf5 complex members are shared between *Leishmania* and *T. brucei*, others are not.

The findings here will serve as an excellent jumping off point for the further characterization of other bromodomain proteins in *Leishmania*, while also making important contributions to the understanding of how gene regulatory mechanisms operate in this early branching organism. Thus, the paper is appropriate to the wide readership offered by Nature Communications.

In general, the experiments presented in the paper are technically sound and are very well controlled. I especially appreciated the add backs of the wildtype and mutant versions of Bdf5 and the use of the diCre system. I have only a few major suggestions and a number of smaller points that I'm hoping will not be too onerous in revision.

Major Points

I may have missed it, but I don't think the ChIP-seq experiments included a control in the form of a pulldown with a nonspecific (IgG) antibody or an HA pulldown in an untagged parasite line. It would be nice to see one or the other of these controls, though certainly both are not necessary. At least in my hands, some peaks are called by MACS when comparing the IP material to the input control, but these same peaks appear in the IgG or the HA control pulldowns. Particularly because Bdf5 showed up in some rather surprising regions, it would be good to verify that these peaks of localization are genuine and not some artifact of sticky DNA or something else. If the authors have performed lots of ChIP-seq using this same workflow and have such a control on hand that would be ideal. Otherwise it might make sense to perform at least one replicate of one of these controls to validate all the called peaks in the experimental samples. Bedtools can easily be used to filter out peaks that appear in the control IPs.

This is an important point which we addressed by performing a control ChIP-seq experiment using the same anti-HA magnetic beads only this time against the parental Lmx DiCre strain that did not contain a tagged version of BDF5. The results indicated that no regions of the genome that were enriched by BDF5 ChIP were enriched in the control ChIP. The fastq files for the raw data will be deposited at ENA under the same project number and also at TriTrypDB. Control tracks have been added to Figure 4.

Figure 3. The visualization for Bdf5 localization in all the *Leishmania* chromosomes is neat, but there's so much going on that I think the main points about the localization for Bdf5 are not easy to see using this visualization. The authors might consider swapping in Figure S7 as the main figure in the paper while putting the current Figure 3 as the supplement.

*We agree that the circus plot is a dense figure, however we still think it is important to show the global context of BDF5 enrichment. However, we have moved Figure S7 into the main figures as new figure Figure 5 as this gives a very clear example of the BDF5 enrichment at loci on an individual chromosome, which may also be indicative to a general reader the organisation of a typical *Leishmania* chromosome. It also includes tracks for the control ChIPseq data.*

The experiments presented in Figure S11 and S12 are really hard to understand without some description of the

assay in the text. If length constraints are preventing the authors from including this in the main text, it would be great to include it as supplemental text somewhere. I think the result that Bdf5 ablation doesn't affect splicing is an important one, and is given rather short shrift as presented here.

We apologise for the lack of clarity here and have expanded the description of this experiment with the addition of this paragraph:

*"Despite enrichment in the BDF5 proximal proteome for mRNA splicing factors, we did not find evidence to support trans- or cis-splicing defects in BDF5 induced-null mutants using a qualitative RT-PCR assay. A positive control a strain of *L. mexicana* was generated using CRISPR/Cas9 precision editing of CRK9 at the codon encoding the M501 gatekeeper residue to glycine codon (Supplementary Fig. 10). This mutant is specifically inhibited by the bumped kinase inhibitor 1NM-PP1 leading to defects in splicing. Cis-splicing of poly-A polymerase and trans-splicing of cyclophilin A mRNA was examined by an RT-PCR method that could detect the pre-mRNA and mature mRNA. Deletion of BDF5 from cells did not cause changes in the abundance of the mature mRNAs but CRK9M501G inhibition did result in accumulation of unspliced pre-mRNA (Supplementary Fig. 11)."*

Minor Points.

1. Something I found slightly surprising was that in the result for Figure 2E, the median levels of parasite burden are 10-fold lower in the DMSO treated Bdf5-/+flx pNUSBdf5 compared to the DMSO treated Bdf5+/+flx condition. I noticed the same thing for the result presented in Figure S6C. I think this difference was mentioned in the text but could the authors speculate a bit more on what's going on here? Is the difference statistically significant? Is there some leaky flipping of the pNUS allele (doesn't seem so from the gels) or is there some other explanation the authors could postulate?

When we expand the multiple comparisons here there is no statistically significant decrease, so it may just be due to random variation. It does look like the median is skewed by one mouse that had a much lower burden of parasites, which could occur from natural variability in the infection progression or during the injection of the footpads.

If the slight difference is intrinsic to the parasite strain it could potentially be due to fine-tuning of the levels of BDF5. The BDF5-/+ being a heterozygote, and BDF5-/+ pNUS BDF5 being a "homozygote" but with variable levels of episome in cells within the population. We should also make clear that in this experiment the pNUS episome being used generates constitutive transcription, and is not induced by flipping loxP sites, as is performed later with integrated pRIB versions of BDF5^{wt} and BDF5^{N9F/N257F}.

2. Line 239, I think the figure callout should be S3A, S3B rather than S2A, S2B.

Corrected

3. In the paragraph starting on line 264, the authors discuss that the Bdf5 add back doesn't fully complement. Could it be that the addback is missing some regulatory mechanism that exists for the endogenous version? Might it be possible to speculate on why the complementation was not complete? If not, that's ok, I realize some results are just puzzling and we occasionally have to accept that.

In this instance the addback was expressed from a pNUS episome that does not integrate into the genome, therefore individual parasites can stochastically lose the episome, and in this case die. Although this makes the complementation partial in the clonal survival assay, it still reflects our interpretation that BDF5 is essential.

4. Line 92 should read transcriptional start sites rather than transcriptionally start sites.

Corrected

5. In the section starting at line 324, can the authors be a little more explicit about the controls used to call peaks (input samples) and how peaks were specifically called (MACS?).

We apologise for this confusion, the enrichment calling was actually done using this using Deeptools' bamCompare (version 3.3.1) with SES normalisation and bin size of 500 to give log2 fold enrichment tracks from the 3 replicate experimnts. These were converted to .wig files which were combined to give the mean log2 fold enrichments (custom python script combine_wigs.py), the file was filtered for peaks by only including bins with value >0.5 (filter_wig.py) and then merged peaks that were less than 5kb from each other (merge_wig_peaks.py). We have expanded the methods section to make this clear and the python code has been made available as a separate document.

We have performed a series of control ChIPseq experiments on the parental DiCre strain, which yielded no peaks overlapping with BDF5 enriched loci.

6. I think there may be a stray 'which' in line 387, or there is something weird going on with that sentence.

Corrected

7. I'm so impressed with all the co-IPs conducted. Since so many produced nice results, would it be possible to include something in the text such as 'of the x interactions we tested were able to verify Y of them.'

Included in the results section: "Of the 22 proteins (excluding CLK2) we tested under crosslinking conditions 19 were found to co-precipitate BDF5."

8. In the co-IP S8 Figure, would it be possible to put the predicted size in parentheses next to the label for each protein tested? Since many co-IPs produced multiple bands on the gel for the bait protein, it's sometimes difficult to ascertain which band is the relevant one to be looking at. In addition, some of the bands for Bdf5 are rather faint. It would be helpful to indicate which interactions the authors deemed verified by including a star or a plus sample by the relevant lanes.

The predicted mw has been added below the relevant lane and an asterisk has been added for proteins co-precipitating BDF5.

9. Figure 6B, The label for chromosome 8 has a typo.

This was to reflect a fusion of Chr8 and 29 which is found in L. mexicana compared to L. major, but we have removed this for clarity.

10. In the ChIP-seq methods section, could just a little more detail be included for how these samples were analyzed? For example, how many mismatches were allowed during alignment, and were reads aligned uniquely or not? One figure legend mentions using MACS to call peaks but I don't see that represented here. If MACS was used what mode was run: broad? Narrow?

We apologise for this confusion, the enrichment calling was actually done using this using Deeptools' bamCompare (version 3.3.1) with SES normalisation and bin size of 500 to give log2 fold enrichment tracks from the 3 replicate experiments. These were converted to .wig files which were combined to give the mean log2 fold enrichments (custom python script combine_wigs.py), the file was filtered for peaks by only including bins with value >0.5 (filter_wig.py) and then merged peaks that were less than 5kb from each other (merge_wig_peaks.py). We have expanded the methods section to make this clear and the python code has been made available as a separate document..

11. I can't find any mention of Figure S3 in the text, as noted again below in the Figure legend section. Are the G1, S, and G2 phases gated here?

The cell cycle data was re-analysed using FCS Express 7 Flow version 7.10.0007 (De Novo Software, Inc.) using the Multicycle DNA content feature to better estimate the proportion of cells in each stage of the cell cycle. This has been represented in an updated Supplemental Figure 3.

12. In general, the figure legends could benefit with some revision so that they can stand alone without having to refer to the text. I think all the figured legends should be carefully looked through, but I'll cite some examples of below.

We have added extra detail to figure legends to help guide readers through them. Specific examples are addressed below.

Examples for figure legends:

Figure 2A should include a description of how parasites were treated for 48 hours, then diluted, then treated for the remaining time, etc.

The legend has been adjusted: "Growth curve of promastigotes treated with the inducing agent, rapamycin (Rap.), or the vehicle, DMSO. For the first 48 h 300 nM rapamycin was added. At 48h the cultures were passaged, and the concentration of rapamycin was lowered to 100 nM. Daily counting was conducted of triplicate cultures, of two independent clones, using a haemocytometer."

Figure 2C legend could be replaced with 'Western blot showing levels of BDF5::6XHA protein following treatment with either DMSO or Rapamycin for 72 hours. The 3 samples shown represent biological replicates.' (Are these biological replicates, I wasn't sure???)

The legend has been changed to include: "Western blot showing levels of BDF5::6xHA protein after 72 h of DMSO or rapamycin treatment, conducted in biological triplicate."

Figure 2E should mention that parasites were pretreated with DMSO or Rapamycin and then used to infect mice, etc.

The legend has been altered to contain: "Late-log cultures were pre-treated with 300 nM rapamycin and allowed to become stationary, prior to footpad infection for 8-weeks."

Figure S1A could use a better description of the cartoon. All the cartoons work really well and their inclusion is much appreciated!

An updated legend has been added to better explain the CRISPR/Cas9 workflow.

Figure S2 legend should define TCM and TCK as PDB IDs.

This has been added to the legend.

Figure S3. Are these biological replicates? Are the results quantified somewhere? I can't find any reference to Figure S3 in the text, though maybe I missed it.

Referrals to Fig S3 have been added, these data are from a single representative biological replicate. The cell death phenotype is supported by the clonal survival assays which were conducted in triplicate (Fig. 2D).

Figure S5. It wasn't clear to me if the pRIB construct was randomly integrated or targeted somewhere into the Lmx:DiCre line.

We have added "After integrating this into rRNA locus of Lmx::DiCre," to make it clear that this construct is targeted to the rRNA locus.

Figure S7. I'm not sure what the logEvsI track represents, please include in legend.

This is the log₂ fold enrichment of the elution fraction over the input fraction, the figure has been adjusted to make this clear.

Figure S9. Missing legend for panel E.

Legend added.

Reviewer #3 (Remarks to the Author):

The manuscript by Jones et al. examines bromodomain factors (BDF) of Leishmania. They present though description into the identification of BDF containing proteins and whether they are required for viability. They go on to provide extensive functional characterization of BDF5. Overall, this is a really nice manuscript that is technically sound and I believe will be of broad interest.

Minor Comments:

- Figure 4 is overall a nice summary of the BioID results. One minor point is that the Dash Outline denoting Co-IP verified is hard to see.

The outlines have been changed to a thick bold outline to be more easily visible.

- Supplemental datasets of BioID data would benefit from a brief legend describing the columns contained in each file.

A legend was added to the BioID spreadsheets to explain the columns

- miniTurboID experiments are not mentioned in the methods.

We have added a comprehensive methods section for the XL/BioID/miniTurbo ID experiments.

- More complete methods for BioID including mass spectrometry details, number of replicates, and statistical analysis to determine enrichment of BDF5 over control would be nice.

We have updated this section to provide a more comprehensive method of the BioID methods, mass spec details, replicates and statistical analysis.

- Likely nothing further is needed for this manuscript but the phosphoproteomic miniTurboID data sampled over time seems under explored.

We have changed the discussion section to more specifically consider the potential for phosphorylation of BDFs to enable functional regulation. This is exemplified by the structural re-arrangements of human BRD4 conferred by phosphorylation by CK2 which causes structural rearrangement and allows the BDs to engage with acetylated peptides. I hope to be able to explore BDF5 functional regulation by phosphorylation in the future.

Other notes: We have changed TSS (Transcriptional Start Site) to TSR (Transcriptional Start Region) to be more consistent with the terminology of Siegel. This reflects that the precise start site of transcription is not well defined and that the regions are quite large, as opposed to discrete sites. A paragraph of introduction describing the role of BDF5s in other kinetoplastids was removed to reduce the length of the manuscript. This section did not contain material of direct relevance to BDF5 that was not repeated in the discussion, so we did not think it alters the scope of the manuscript significantly.

The model of the CRKT complex has been edited to represent nucleosomes as discs rather than spheres.

Referrals in the text to "KKT19" have been replaced with "CLK2" to be consistent with other recent outputs from our laboratory (Saldivia et al. 2020, 2021. PMID: 34128702, PMID: 32661312).

REVIEWERS' COMMENTS

Reviewer #1 (Remarks to the Author):

The revised manuscript is much improved, and addresses all concerns raised. I congratulate the authors on the thoroughness of their experimentation, the large quantum of work done, and their exciting findings. The data will add a new dimension to the poorly understood Leishmania transcriptional process.

This manuscript can definitely be published in its present form.

Reviewer #2 (Remarks to the Author):

The authors have done an excellent job of addressing all the points made by the reviewers. I'm excited to see this work published!

One minor thing is that a sentence included in the rebuttal looks like it contains a stray 'indicating' word:

The proportion of non-viable cells in the cultures at this point was ~10% (Fig. S3D) indicating and in combination with other experiments indicates that BDF5 depletion leads to a rapid cytostatic phenotype followed by eventual cell death.”

We again thank all the reviewers for their efforts to give a constructive critique of our work.

REVIEWERS' COMMENTS

Reviewer #1 (Remarks to the Author):

The revised manuscript is much improved, and addresses all concerns raised. I congratulate the authors on the thoroughness of their experimentation, the large quantum of work done, and their exciting findings. The data will add a new dimension to the poorly understood Leishmania transcriptional process. This manuscript can definitely be published in its present form.

No further action.

Reviewer #2 (Remarks to the Author):

The authors have done an excellent job of addressing all the points made by the reviewers. I'm excited to see this work published!

One minor thing is that a sentence included in the rebuttal looks like it contains a stray 'indicating' word:

The proportion of non-viable cells in the cultures at this point was ~10% (Fig. S3D) indicating and in combination with other experiments indicates that BDF5 depletion leads to a rapid cytostatic phenotype followed by eventual cell death.”

This has been corrected to: The proportion of non-viable cells in the cultures at this point was ~10% (Supplementary Fig. 3d), in combination with our other experiments this suggests that BDF5 deletion leads to a rapid cytostatic phenotype followed by eventual cell death.